# Partial Identification of Policy Values under Network Interference

**Ziyan Wang** [1]   **Yiran Liu** [3]   **Zhiheng Zhang**[✉ 1 2]

## Abstract

Offline Policy Evaluation (OPE) aims to estimate the value of a target policy from historical logged data without interacting with the environment, thereby assessing policy performance. In settings with network interference, individuals no longer satisfy the SUTVA assumption: an individuals outcome is influenced not only by their own treatment but also by the treatments of their neighbors, which makes the definition and estimation of policy value more complex. To capture this interference mechanism, we allow all neighbors to affect individual outcomes through a unified exposure mapping, and we use a decaying higher-order neighborhood aggregation to characterize the influence of more distant neighbors. Moreover, in real-world applications, the target policy and the logging policy often do not fully overlap (non-overlap), so the policy value in non-overlap regions cannot be point-identified. To address this issue, we partially identify the policy value over non-overlap regions and, under a smoothness assumption, formulate the estimation of the lower and upper bounds as a linear program, yielding valid bounds on the offline policy value. Finally, we conduct systematic experiments on semi-synthetic network data to validate the effectiveness and robustness of the proposed method under network interference and limited overlap.

## 1 Introduction

Evaluating the value of a counterfactual policy from observational data is a central problem in causal inference and decision-making. Given data collected under a behaviour (logging) policy, the goal is to assess the performance of a target policy without deploying it in the environment. This problem underlies policy learning, welfare analysis, and algorithmic decision-making, and has been studied extensively in causal inference, econometrics, and reinforcement learning (Hernán & Robins, 2020; Imbens & Rubin, 2015; Dudík et al., 2011).

Classical approaches rely on unconfoundedness, SUTVA, and overlap between the behavior and target policies. Under these conditions, policy values are point-identified and can be consistently estimated using inverse probability weighting, regression adjustment, or doubly robust methods (Robins & Rotnitzky, 1994; Dudík et al., 2011; Chernozhukov et al., 2018), with notable success in settings with independent units and sufficiently exploratory assignment.

However, two features of modern large-scale decision systems challenge this paradigm. First, overlap may fail: behavior policies can be conservative, deterministic, or operationally constrained, creating regions where the target policy assigns positive probability but the behavior policy assigns none. In such non-overlap regions, policy values are no longer point-identified and standard estimators can be unstable due to extreme propensity weights. This has motivated work on *partial identification* of policy values, which characterizes valid bounds under smoothness or structural assumptions (Manski, 2009; Khan et al., 2024).

Second, interference between units often violates SUTVA. In many systems—including social networks, marketplaces, and spatial environments—an individual's outcome depends on others' treatments, making the potential-outcome framework intrinsically high-dimensional. A growing literature addresses interference via exposure mappings, neighborhood summaries, or design-based restrictions (Hudgens & Halloran, 2008; Aronow & Samii, 2017; Leung, 2022), but most work assumes overlap or relies on randomized (or partially randomized) designs.

Importantly, limited overlap and network interference have largely been studied in isolation. Recent work on off-policy evaluation beyond overlap establishes partial identification results under smoothness, but relies on independence across units (Khan et al., 2024). Conversely, inter-

---

[1]School of Statistics and Data Science, Shanghai University of Finance and Economics, Shanghai, China [2]Institute of Big Data Research, Shanghai University of Finance and Economics, Shanghai, China [3]College of Business, Shanghai University of Finance and Economics, Shanghai, China. Correspondence to: Zhiheng Zhang <zhangzhiheng@mail.shufe.edu.cn>.

*Proceedings of the 43rd International Conference on Machine Learning*, Seoul, South Korea. PMLR 306, 2026. Copyright 2026 by the author(s).

ference frameworks model spillovers but typically assume overlap or experimental variation (Hudgens & Halloran, 2008; Leung, 2022). Thus a basic question remains: *to what extent can policy values be identified when both overlap and SUTVA fail?*

We address this question by developing a theory of *partial identification of policy values under network interference*. Rather than point estimation, we characterize bounds consistent with the data and interpretable structural assumptions. We formalize interference through a truncated exposure mapping that aggregates treatments within a finite neighborhood, allowing higher-order spillovers to decay with graph distance. Under an approximate neighborhood interference condition (Leung, 2022), this reduces dependence on the global assignment vector to a low-dimensional exposure state, up to a controlled truncation bias. Building on this representation, we decompose the policy value into overlap and non-overlap components: the overlap part is estimable with existing methods, while the non-overlap part is only partially identified. We show that a graph-augmented smoothness condition—coupling covariates, exposure summaries, and network distance—yields *computable bounds* for the non-overlap policy value. The bounds are characterized by a linear program with a closed-form solution; their tightness depends on network geometry and the density of overlap. Moreover, the approach remains robust to network misspecification.

Our results subsume existing partial identification results without interference as a special case and expand the settings where informative policy evaluation is possible. Our contributions are:

- We study policy evaluation under simultaneous violations of overlap and SUTVA, establishing partial identification of policy values in networks.

- We introduce a graph-augmented smoothness condition under which the non-overlap component admits tight, computable bounds with closed-form solutions.

- We evaluate the resulting intervals on a semi-synthetic dataset: across truncation levels and smoothness parameters, the estimated intervals achieve near-nominal coverage while remaining relatively tight.

## 2 Model and Notation

We observe a population of $n$ units indexed by $\mathcal{N} := \{1, \ldots, n\}$ connected by a (possibly weighted) graph $\mathcal{G} = (\mathcal{N}, \mathcal{E})$ with adjacency matrix $A \in \{0,1\}^{n \times n}$.[1] Let $X_i \in$

---

[1] For simplicity we present the binary, undirected case with $A_{ii} = 0$ and $A_{ij} = A_{ji}$. All definitions extend to directed or weighted graphs by replacing graph distance and neighborhood operators accordingly.

$\mathcal{X}$ denote the observed covariates of node $i$, and write $\mathbf{X} := (X_1, \ldots, X_n)$. Let $\text{dist}_{\mathcal{G}}(i, j)$ be a graph distance (e.g., shortest-path distance). For $s \geq 1$, define the $s$-hop neighborhood

$$\mathcal{N}_s(i) := \{j \in \mathcal{N} : \text{dist}_{\mathcal{G}}(i, j) = s\}.$$

We consider binary treatment assignments $\mathbf{D} \in \{0,1\}^n$, where $D_i$ is the treatment of node $i$. A (possibly stochastic) *policy* $\pi$ is a conditional distribution over assignment vectors given the observed network, i.e.,

$$\mathbf{D}^\pi \sim \pi(\cdot \mid \mathbf{X}, A), \ \pi : \{0,1\}^n \times (\mathcal{X}^n \times \{0,1\}^{n \times n}) \to [0,1].$$

We write $\pi_b$ for the *behavior/logging* policy that generates the observed data and $\pi_e$ for the *evaluation* policy whose value we aim to assess. The observed dataset consists of a single network realization

$$(A, \mathbf{X}, \mathbf{D}, \mathbf{Y}) \quad \text{with} \quad \mathbf{D} \sim \pi_b(\cdot \mid \mathbf{X}, A), \ Y_i = Y_i(\mathbf{D}),$$

where $\mathbf{Y} := (Y_1, \ldots, Y_n)$ and $Y_i(\mathbf{D})$ denotes the realized outcome under the realized assignment vector. This formulation allows *arbitrary interference*: outcomes may depend on the entire assignment vector, as in the network-interference literature (Hudgens & Halloran, 2008; Aronow & Samii, 2017).

For each node $i$ and each assignment vector $\mathbf{d} \in \{0,1\}^n$, let $Y_i(\mathbf{d}) \in \mathbb{R}$ denote the potential outcome of node $i$ under $\mathbf{D} = \mathbf{d}$. Equivalently, one may view the data as generated from a structural model (Ma & Tresp, 2021) in which the assignment mechanism is governed by $\pi_b$ and the outcome mechanism permits dependence on $(\mathbf{d}, \mathbf{X}, A)$:

$$\mathbf{D} \sim \pi_b(\cdot \mid \mathbf{X}, A), \tag{1}$$

$$Y_i = f_Y(i, \mathbf{D}, \mathbf{X}, A) + \varepsilon_i, \qquad i \in \mathcal{N}, \tag{2}$$

with $\mathbb{E}[\varepsilon_i \mid \mathbf{X}, A, \mathbf{D}] = 0$. No independence across $i$ is assumed a priori; dependence is mediated through $(\mathbf{D}, A)$. A direct analysis of $Y_i(\mathbf{d})$ is generally intractable because $\mathbf{d}$ is $n$-dimensional. Following the exposure mapping paradigm, we summarize the relevant aspects of $\mathbf{d}$ for node $i$ via a deterministic mapping

$$T_i(\mathbf{d}) := \big(d_i, \ E_i(\mathbf{d}; A)\big),$$

where $E_i$ aggregates spillovers from other nodes. In this paper we focus on a decaying higher-order neighborhood aggregation $E_i(\mathbf{d}; A) := \sum_{s \geq 1} w_s U_i^{(s)}(\mathbf{d})$, where

$$U_i^{(s)}(\mathbf{d}) := \frac{1}{|\mathcal{N}_s(i)|} \sum_{j \in \mathcal{N}_s(i)} d_j. \tag{3}$$

Here $(w_s)_{s \geq 1}$ are nonnegative weights with $w_{s+1} \leq w_s$ (and typically $\sum_{s \geq 1} w_s = 1$). Thus $U_i^{(s)}(\mathbf{d})$ is the fraction of treated $s$-hop neighbors and $w_s$ downweights more

distant interference. When needed for computation or identifiability, we will work with an $S$-truncated exposure $E_i^{(S)}(\mathbf{d}) := \sum_{s=1}^{S} w_s U_i^{(s)}(\mathbf{d})$ and the corresponding truncated exposure state $T_i^{(S)}(\mathbf{d}) := (d_i, E_i^{(S)}(\mathbf{d}))$; the truncation level $S$ plays a central role in controlling approximation bias under approximate neighborhood interference (Leung, 2022).

We adopt a conditional-mean exposure-response restriction: there exist functions $\mu(\cdot)$ such that for every $\mathbf{d} \in \{0,1\}^n$,

$$\mathbb{E}[Y_i(\mathbf{d}) \mid X, A] = \mu(i, T_i(\mathbf{d})), \qquad i \in \mathcal{N}. \quad (4)$$

Importantly, (4) is weaker than assuming $Y_i(\mathbf{d})$ depends on $\mathbf{d}$ *only* through $T_i(\mathbf{d})$ almost surely; it suffices for policy value analysis since our target is an average outcome. Let $\mathcal{T}$ denote the range of the exposure mapping $T_i(\cdot)$ (a subset of $\{0,1\} \times [0,1]$ under (3)). For a policy $\pi$, define the *policy-induced exposure distribution*

$$\pi(i, t) := \mathbb{P}_{\mathbf{D} \sim \pi(\cdot \mid \mathbf{X}, A)}(T_i(\mathbf{D}) = t \mid \mathbf{X}, A), \qquad t \in \mathcal{T}.$$

The (conditional) value of policy $\pi$ is the average expected outcome under the assignment $\mathbf{D}^\pi$:

$$V(\pi \mid \mathbf{X}, A) := \frac{1}{n} \sum_{i \in \mathcal{N}} \mathbb{E}_{\mathbf{D} \sim \pi}[Y_i(\mathbf{D}) \mid \mathbf{X}, A]. \quad (5)$$

By iterated expectation and (4), we obtain the key reduction

$$\begin{aligned}
\mathbb{E}_{\mathbf{D} \sim \pi}[Y_i(\mathbf{D}) \mid \mathbf{X}, A] &= \mathbb{E}_{\mathbf{D} \sim \pi}[\mathbb{E}[Y_i(\mathbf{D}) \mid \mathbf{D}, \mathbf{X}, A] \mid \mathbf{X}, A] \\
&= \mathbb{E}_{\mathbf{D} \sim \pi}[\mu(i, T_i(\mathbf{D})) \mid \mathbf{X}, A] \\
&= \sum_{t \in \mathcal{T}} \mu(i, t)\, \pi(i, t), \quad (6)
\end{aligned}$$

where the sum is understood as an integral when $\mathcal{T}$ is continuous. Substituting (6) into (5) yields

$$V(\pi \mid \mathbf{X}, A) = \frac{1}{n} \sum_{i \in \mathcal{N}} \sum_{t \in \mathcal{T}} \mu(i, t)\, \pi(i, t). \quad (7)$$

Equation (7) rewrites a high-dimensional expectation over assignment vectors as an exposure-level mixture of exposure-specific mean outcomes, and forms the basis for our partial identification analysis.

**Overlap in exposure space and the target problem.** Define the behavior-policy support for node $i$ in the exposure space by

$$\mathcal{O}_i := \{t \in \mathcal{T} : \pi_b(i, t) > 0\}, \qquad \mathcal{O}_i^c := \mathcal{T} \setminus \mathcal{O}_i.$$

When $\pi_e(i, t) > 0$ for some $t \in \mathcal{O}_i^c$, the contribution of such exposure states to (7) cannot be point-identified from logged data without additional structure; this is precisely

the *non-overlap* regime. Our goal is to characterize (and estimate) valid bounds on $V(\pi_e \mid \mathbf{X}, A)$ under network interference and exposure non-overlap.

To make the identification problem in (7) well-posed yet nonparametric, we will invoke three types of structural restrictions later:

*Assumption* 2.1 (Approximate Neighborhood Interference (Leung, 2022)). Let $N_{\leq S}(i) := \{j \in \mathcal{N} : \text{dist}_{\mathcal{G}}(i, j) \leq S\}$, For each unit $i \in \mathcal{N}$ and each integer $S \geq 1$, let $\mathbf{D}^{(i,S)}$ denote a counterfactual assignment vector obtained by resampling $\{D_j : j \in N_{\leq S}(i)\}$ independently from their marginal distribution, while keeping $\{D_j : j \notin N_{\leq S}(i)\}$. fixed.

Define the truncation error $\theta_n(S) := \max_{i \in \mathcal{N}} \mathbb{E}[|Y_i(\mathbf{D}) - Y_i(\mathbf{D}^{(i,S)})| \mid \mathbf{X}, A]$. We assume that $\theta_n(S)$ is non-increasing in $S$ and that there exists a (possibly data-dependent) truncation level $S = S_n$ such that $\theta_n(S_n) \leq \bar{\theta}_n$, $\bar{\theta}_n \to 0$ as $n \to \infty$.

*Assumption* 2.2 (Graph-Augmented Smoothness). Let $T_i^{(S)} = (D_i, E_i^{(S)})$ be the $S$-truncated exposure and define the node representation $Z_i := (X_i, T_i^{(S)})$. Let $d_G(i, j)$ be a graph distance on $\mathcal{G}$ and define the composite metric $d_\lambda(i, j) := \|Z_i - Z_j\|_W + \lambda\, d_G(i, j)$, $\lambda \geq 0$, where $\|\cdot\|_W$ is a weighted norm on the feature-exposure space. We assume that the conditional mean function $\mu(\cdot)$ is $L$-Lipschitz with respect to $d_\lambda$, i.e., $|\mu(i, T_i) - \mu(j, T_j)| \leq L\, d_\lambda(i, j)$, $\forall i, j \in \mathcal{N}$.

*Assumption* 2.3 (Overlap Denseness). Let $\mathcal{O}_i = \{t \in \mathcal{T} : \pi_b(i, t) > 0\}$ be the exposure support of the behavior policy for unit $i$, and define the set of overlap nodes $\mathcal{O}_n := \{i \in \mathcal{N} : \mathcal{O}_i \neq \emptyset\}$. For each $i \in \mathcal{N}$, define the distance to the overlap set $\rho_n(i) := \inf_{j \in \mathcal{O}_n} d_\lambda(i, j)$. We assume that there exists a deterministic sequence $r_n \downarrow 0$ such that $\sup_{i \in \mathcal{N}} \rho_n(i) \leq r_n$ with probability $1 - o(1)$.

**Discussion of assumptions.** Assumptions 2.1–2.3 make the identification problem in (7) well-posed without imposing parametric structure. Assumption 2.1 reduces the effective dimensionality of interference by requiring spillovers to decay with graph distance; it generalizes approximate neighborhood interference (Leung, 2022) and can be probed via the sensitivity of the bounds to the truncation level $S$. Assumption 2.2 imposes Lipschitz smoothness on the *conditional mean* (not individual outcomes), paralleling smoothness-based partial identification without interference (Khan et al., 2024) while incorporating network geometry; larger $L$ weakens the restriction and widens the bounds. Assumption 2.3 prevents vacuous non-identification by requiring non-overlap exposures to be close (under $d_\lambda$) to observed overlap states, which is checkable from the empirical distribution of $(X_i, T_i^{(S)})$. Weakening any assumption-larger $L$, smaller $S$, or larger

$r_n$ monotonically expands the identification region, yielding a built-in sensitivity analysis. All three are essential: truncation controls dimensionality, smoothness constrains extrapolation, and overlap denseness links observed to unobserved exposure regimes.

In the i.i.d. setting (Khan et al., 2024), the estimand decomposes over actions and overlap is defined in covariate–action space. Under network interference, even with binary treatments, (i) $Y_i(\mathbf{d})$ may depend on the full assignment $\mathbf{d}$, and (ii) overlap is naturally defined in the *policy-and graph-induced exposure space* via $\pi_b(i, t)$, where $t$ depends on $i$'s neighborhood. These features require combining interference reduction (exposure truncation) with smoothness over a graph-aware metric, developed next.

## 3  Framework

This section develops valid, assumption-transparent bounds for the evaluation-policy value in a network with interference and limited overlap. The organization parallels the non-overlap i.i.d. theory of Khan et al. (2024), but the network setting introduces two *endogenous* challenges: (i) the counterfactual depends on the high-dimensional global assignment vector $\mathbf{D}$, requiring an interference-reduction step with quantifiable truncation bias; and (ii) the sample is not i.i.d. across units, so consistency and rates require controlling dependence induced by the graph and exposure construction. Throughout, $(A, \mathbf{X})$ is treated as given, and probabilities/expectations are with respect to $\mathbf{D} \sim \pi_b(\cdot \mid \mathbf{X}, A)$ and outcome noise unless stated otherwise.

**Reducing global interference via truncation**  Under interference, $Y_i(\mathbf{d})$ may depend on the entire assignment vector $\mathbf{d} \in \{0,1\}^n$, making policy evaluation high-dimensional. We therefore work with a truncated exposure representation and explicitly track the resulting approximation error.

Let $T_i^{(S)}(\mathbf{D}) = (D_i, E_i^{(S)}(\mathbf{D}))$ denote the $S$-truncated exposure state (defined in Section 2). We impose an exposure-response restriction at the level of conditional means.

*Assumption* 3.1 ($S$-order exposure sufficiency for conditional means). There exist functions $\mu(\cdot)$ such that for all assignment vectors $\mathbf{d}$, $\mathbb{E}[Y_i(\mathbf{d}) \mid \mathbf{X}, A] = \mu(i, T_i^{(S)}(\mathbf{d}))$

Assumptions 2.1–3.1 are standard in spirit: ANI formalizes that higher-order interference decays with graph distance (Leung, 2022), while Assumption 3.1 is an exposure mapping restriction (weaker than almost-sure sufficiency) that is natural when the estimand is an *average* outcome.

**Lemma 3.2** (Reduction to a truncated policy value). *Under Assumptions 2.1–3.1, for any policy* $\pi$, $V(\pi \mid \mathbf{X}, A) = V^{(S)}(\pi \mid \mathbf{X}, A) + R_S(\pi)$, $|R_S(\pi)| \leq \theta_n(S)$, *where* $V^{(S)}(\pi \mid \mathbf{X}, A) := \frac{1}{n} \sum_{i=1}^n \mathbb{E}_{\mathbf{D} \sim \pi}\left[ \mu(i, T_i^{(S)}(\mathbf{D})) \,\middle|\, \mathbf{X}, A \right]$.

Lemma 3.2 makes explicit that identification and estimation can focus on $V^{(S)}$; the original target $V(\pi)$ is recovered by widening bounds by $\theta_n(S)$.

**Exposure-wise decomposition and the generic consistency**  Because $T_i^{(S)}$ typically includes a continuous component, we discretize for tractability. Let $\mathcal{E} = \{e_1, \ldots, e_K\}$ be a user-chosen grid and let $Q : [0,1] \to \mathcal{E}$ be a binning map. This step may introduce a binning bias, since exposure values within the same bin are treated as equivalent. Finer grids reduce this approximation error but enlarge the exposure state space and may exacerbate sparsity/non-overlap; coarser grids improve stability at the cost of additional approximation bias.

**Binning bias.**  Discretizing the continuous exposure summary introduces a bias–overlap tradeoff. Under $L$-Lipschitz smoothness in the exposure coordinate, if $r_K$ denotes the maximal within-bin radius, then the continuous policy value $V(\pi)$ and its $K$-bin discretized counterpart $V_K(\pi)$ satisfy

$$|V(\pi) - V_K(\pi)| \leq L r_K.$$

Therefore, discretization bias decreases as the bin radius shrinks. However, finer bins also split the data into more exposure states and may weaken local overlap. Appendix B.10 provides the proof and diagnostic experiments.

Define the discretized exposure $\widetilde{T}_i^{(S)} := (D_i, Q(E_i^{(S)}))$ and its finite support $\mathcal{T}^{(S)} := \{0,1\} \times \mathcal{E}$. For notational simplicity we drop tildes henceforth and treat $T_i^{(S)} \in \mathcal{T}^{(S)}$.

For each $t \in \mathcal{T}^{(S)}$, define the policy-induced exposure probabilities $\pi_\ell(i, t) := \mathbb{P}_{\mathbf{D} \sim \pi_\ell(\cdot \mid \mathbf{X}, A)}\left( T_i^{(S)}(\mathbf{D}) = t \,\middle|\, \mathbf{X}, A \right)$, $\ell \in \{b, e\}$. and we have exposure-specific conditional mean $\mu(i, t) = \mathbb{E}\left[ Y_i(\mathbf{D}) \,\middle|\, T_i^{(S)}(\mathbf{D}) = t, \mathbf{X}, A \right]$, $i \in \mathcal{N}$, By iterated expectation (cf. (7)), the truncated value decomposes as

$$V^{(S)}(\pi_e \mid \mathbf{X}, A) = \frac{1}{n} \sum_{i=1}^n \sum_{t \in \mathcal{T}^{(S)}} \mu(i, t)\, \pi_e(i, t). \quad (8)$$

For each $t \in \mathcal{T}^{(S)}$, define the (node-wise) overlap set and its complement

$$\mathcal{O}_t := \{i \in \mathcal{N} : \pi_b(i, t) > 0\}, \mathcal{O}_t^c := \{i \in \mathcal{N} : \pi_b(i, t) = 0\},$$

and write $\mathbf{1}_{i,t} := \mathbf{1}\{i \in \mathcal{O}_t^c\}$. Using (8), we decompose

$$V^{(S)}(\pi_e \mid \mathbf{X}, A) = V_1^{(S)}(\pi_e \mid \mathbf{X}, A) + V_2^{(S)}(\pi_e \mid \mathbf{X}, A), \quad (9)$$

where

$$V_1^{(S)}(\pi_e \mid \mathbf{X}, A) := \frac{1}{n} \sum_{i=1}^n \sum_{t \in \mathcal{T}^{(S)}} \mu(i,t)\, \pi_e(i,t)\, \mathbf{1}\{i \in \mathcal{O}_t\}, \tag{10}$$

$$V_2^{(S)}(\pi_e \mid \mathbf{X}, A) := \frac{1}{n} \sum_{i=1}^n \sum_{t \in \mathcal{T}^{(S)}} \mu(i,t)\, \pi_e(i,t)\, \mathbf{1}_{i,t}. \tag{11}$$

The term $V_1^{(S)}$ is identifiable from logged data (in principle) since it only involves exposure states observed under $\pi_b$, whereas $V_2^{(S)}$ is not point-identified when $\pi_e(i,t) > 0$ but $\pi_b(i,t) = 0$.

*Remark* 3.3 (Empty overlap buckets). For a fixed exposure state $t$, the closed-form Lipschitz bounds require $\mathcal{O}_t \neq \emptyset$. If $\mathcal{O}_t = \emptyset$ while $\sum_i \pi_e(i,t) > 0$, then there is no observed anchor point for that state, and Lipschitz smoothness alone cannot yield informative bounds. This is an extreme violation of the $t$-specific overlap-denseness condition. In such cases, one must either merge exposure bins to restore support or use only trivial bounds based on a known outcome range. We report empty-bucket and zero-overlap diagnostics in the Table 4.

We state a network-adapted analogue of Khan et al. (2024, Theorem 4.1). The statement is purely in terms of consistency of component estimators; it does not assume i.i.d. sampling and therefore carries over to the network setting.

**Theorem 3.4** (Consistency of partial identification intervals). *Under Assumption 2.1 to 2.3, fix $(\mathbf{X}, A)$ and truncation level $S$. Suppose $\widehat{V}_1^{(S)}$ is a consistent estimator of $V_1^{(S)}(\pi_e \mid \mathbf{X}, A)$ and $\widehat{V}_2^{(S),-}, \widehat{V}_2^{(S),+}$ are consistent estimators of deterministic bounds $V_2^{(S),-} \leq V_2^{(S)} \leq V_2^{(S),+}$. Define $\widehat{V}^{(S),-} := \widehat{V}_1^{(S)} + \widehat{V}_2^{(S),-}$ and $\widehat{V}^{(S),+} := \widehat{V}_1^{(S)} + \widehat{V}_2^{(S),+}$. Then for any $\varepsilon > 0$, $\lim_{n \to \infty} \mathbb{P}\left( \widehat{V}^{(S),-} - \varepsilon \leq V^{(S)}(\pi_e \mid \mathbf{X}, A) \leq \widehat{V}^{(S),+} + \varepsilon \right) = 1$. Consequently, combining with Lemma 3.2 yields a valid asymptotic interval $V(\pi_e \mid \mathbf{X}, A) \in \left[ \widehat{V}^{(S),-} - \bar{\theta}, \ \widehat{V}^{(S),+} + \bar{\theta} \right]$.*

Theorem 3.4 clarifies the proof strategy: (i) estimate the overlap term $V_1^{(S)}$ by any consistent procedure (e.g., IPW/DR with flexible outcome regression), and (ii) construct consistent estimators of valid bounds for the non-overlap term $V_2^{(S)}$.

**Valid bounds under graph-augmented Lipschitz smoothness** We now develop valid bounds for $V_2^{(S)}$ under a nonparametric smoothness restriction that incorporates both covariates and network geometry.

Fix $t \in \mathcal{T}^{(S)}$ and define the non-overlap contribution at $t$ as in (11). Under Assumption 2.2, the valid lower bound

on $V_{2,t}^{(S)}$ is obtained by minimizing over all $L$-Lipschitz extensions that match the true mean on the overlap set: In practice, $\mu(i,t)$ is unknown even in the overlap region and must be estimated. Let $\widehat{\mu}(i,t)$ be any estimator for $i \in \mathcal{O}_t$ based only on overlap data (e.g., a regression/GNN fit restricted to $\pi_b(i,t) > 0$).Since $t$ is fixed, we use $\widehat{\mu}_t(i)$ instead of $\mu(i,t)$ to represent it. Consider the linear program $\widehat{V}_{2,t}^{(S),-} :=$

$$\begin{aligned} \min_{u_1,\ldots,u_n} \quad & \frac{1}{n} \sum_{i=1}^n u_i\, \pi_e(i,t)\, \mathbf{1}_{i,t} \\ \text{s.t.} \quad & |u_i - u_j| \ \leq \ L\, d_\lambda(i,j),\ 1 \leq i < j \leq n, \\ & u_i - \widehat{\mu}_t(i) = 0, \qquad \forall\, i \in \mathcal{O}_t. \end{aligned} \tag{12}$$

*Assumption* 3.5 (Feasibility on the overlap set). For each $t \in \mathcal{T}^{(S)}$, the fitted values $\widehat{\mu}_t(\cdot)$ satisfy $|\widehat{\mu}_t(i) - \widehat{\mu}_t(j)| \leq L\, d_\lambda(i,j), \qquad \forall i,j \in \mathcal{O}_t$.

Assumption 3.5 is the network analogue of Khan et al. (2024); it is necessary for (12) to be feasible. It can be diagnosed empirically by checking the Lipschitz inequalities on observed overlap pairs. The next theorem is the network analogue of Khan et al. (2024, Theorem 4.5). The proof follows the same "no-interaction" logic, but now hinges on the fact that $d_\lambda$ remains a metric despite incorporating graph structure.

**Theorem 3.6** (valid bounds and closed-form solutions under graph-Lipschitz smoothness). *Fix $t \in \mathcal{T}^{(S)}$ and suppose Assumption 2.2 holds.*

1. *Under Assumption 3.5, the program (12) is feasible and its optimal value admits the closed form*

$$\widehat{V}_{2,t}^{(S),-} = \frac{1}{n} \sum_{i=1}^n \pi_e(i,t) \left( \max_{j \in \mathcal{O}_t} \widehat{\mu}_t(j) - L\, d_\lambda(i,j) \right) \mathbf{1}_{i,t}. \tag{13}$$

*Analogously, the upper bound equals*

$$\widehat{V}_{2,t}^{(S),+} = \frac{1}{n} \sum_{i=1}^n \pi_e(i,t) \left( \min_{j \in \mathcal{O}_t} \widehat{\mu}_t(j) + L\, d_\lambda(i,j) \right) \mathbf{1}_{i,t}.$$

2. *The population sharp lower bound in (12) equals*

$$V_{2,t}^{(S),-} = \frac{1}{n} \sum_{i=1}^n \pi_e(i,t) \left( \sup_{j \in \mathcal{O}_t} \mu_t(j) - L\, d_\lambda(i,j) \right) \mathbf{1}_{i,t}, \tag{14}$$

*(with $\sup = \max$ since $\mathcal{O}_t$ is finite), and this bound is valid in the sense that it is attained by the $L$-Lipschitz extension $u_i^\star = \sup_{j \in \mathcal{O}_t} \mu_t(j) - L d_\lambda(i,j)$. An analogous statement holds for $V_{2,t}^{(S),+}$.*

3. *Assume, in addition, that $\max_{i \in \mathcal{O}_t} \left| \widehat{\mu}_t(i) - \mu_t(i) \right| \xrightarrow{\mathbb{P}} 0$, and that the overlap set is dense in the metric $d_\lambda$ in*

*the sense that $r_{n,t} := \max_{i \in \mathcal{O}_t^c} \min_{j \in \mathcal{O}_t} d_\lambda(i,j) \xrightarrow{\mathbb{P}}$ 0. Then $\widehat{V}_{2,t}^{(S),-} \xrightarrow{\mathbb{P}} V_{2,t}^{(S),-}$ and $\widehat{V}_{2,t}^{(S),+} \xrightarrow{\mathbb{P}} V_{2,t}^{(S),+}$.*

Theorem 3.6 captures the key identification insight: even when $\pi_b(i,t) = 0$ (no overlap), smoothness in the *graph-augmented* metric transfers information from overlap nodes to non-overlap nodes, yielding the strongest bounds compatible with the assumptions. A natural skepticism is that the closed-form expressions in (13)–(14) resemble the no-interference results of Khan et al. (2024). This similarity is deliberate: we show that *sharpness (fixed t) and computability are preserved* even when the data are generated on a network with interference and limited overlap. Crucially, the network setting introduces three difficulties absent in the i.i.d. case. First, under interference the policy value depends on the *entire* assignment vector, so we reduce the estimand to a truncated exposure representation with explicit approximation error $\theta_n(S)$ (Lemma 3.2). Second, overlap is no longer defined in a covariate–action space: it is determined by the policy-induced exposure support $\{\pi_b(i,t) > 0\}$, where $t = T_i^{(S)}(\mathbf{D})$ depends on the network $A$ and joint assignment $\mathbf{D}$. Third, statistical guarantees require controlling network dependence and graph geometry. These enter through (i) the graph-augmented metric $d_\lambda$ (which must remain a metric to retain the no-interaction property) and (ii) the overlap covering radius $r_{n,t}$ and effective sample size $n_{\text{eff}}$ in Section 4. Thus, while the closed form mirrors Khan et al. (2024) algebraically, its validity in networks hinges on new interference-reduction and graph-geometry arguments.

Summing Theorem 3.6 over $t \in \mathcal{T}^{(S)}$ yields bounds for $V_2^{(S)}$: $\widehat{V}_2^{(S),\pm} := \sum_{t \in \mathcal{T}^{(S)}} \widehat{V}_{2,t}^{(S),\pm}$, $V_2^{(S),\pm} := \sum_{t \in \mathcal{T}^{(S)}} V_{2,t}^{(S),\pm}$. Combining with any consistent $\widehat{V}_1^{(S)}$ in Theorem 3.4 gives an interval for $V^{(S)}(\pi_e \mid \mathbf{X}, A)$, and Lemma 3.2 lifts it to an interval for $V(\pi_e \mid \mathbf{X}, A)$ by adding $\pm\bar{\theta}$. Algorithm 1 summarizes the procedure and supports sensitivity analyses in $S$ and $L$. Increasing $S$ reduces truncation bias but enlarges the exposure state space; increasing $L$ weakens smoothness and widens bounds.

## 4 Theoretical Analysis

We now characterize how quickly the proposed bound estimators converge and whether these rates are optimal. Relative to Khan et al. (2024), the principal new difficulty is that network dependence reduces the effective amount of information available from $n$ nodes. We capture this via an *effective sample size* $n_{\text{eff}}$, which can be taken as $n$ in i.i.d. special cases and decreases as dependence strengthens.

**Finite-sample error decomposition** Fix $t \in \mathcal{T}^{(S)}$ and de-

---

**Algorithm 1** Partial identification of $V(\pi_e)$ under network interference and non-overlap

---

**Require:** Logged network data $(A, \mathbf{X}, \mathbf{D}, \mathbf{Y})$ under $\pi_b$; evaluation policy $\pi_e$; truncation level $S$; grid $\mathcal{E}$; metric $d_\lambda$; smoothness constant $L$; truncation bound $\bar{\theta}$.

1: Compute $S$-truncated exposures $T_i^{(S)} = (D_i, Q(E_i^{(S)}))$ for all $i$.

2: For each exposure state $t \in \mathcal{T}^{(S)}$, form overlap set $\mathcal{O}_t = \{i : \pi_b(i,t) > 0\}$ and indicator $\mathbf{1}_{i,t} = \mathbf{1}\{i \notin \mathcal{O}_t\}$.

3: Estimate the identifiable overlap value $\widehat{V}_1^{(S)}$ using any consistent OPE estimator restricted to $\{(i,t) : i \in \mathcal{O}_t\}$.

4: **for** each $t \in \mathcal{T}^{(S)}$ **do**

5:     Fit $\widehat{\mu}_t(i)$ for $i \in \mathcal{O}_t$ (e.g., regression/GNN using overlap data only).

6:     Check feasibility on overlap pairs (Assumption 3.5); if violated, increase $L$ or smooth the fit.

7:     Compute $\widehat{V}_{2,t}^{(S),-}$ and $\widehat{V}_{2,t}^{(S),+}$ via the closed forms in (13).

8: **end for**

9: Output interval: $\left[\widehat{V}^{(S),-} - \bar{\theta}, \ \widehat{V}^{(S),+} + \bar{\theta}\right]$, where $\widehat{V}_2^{(S),\pm} := \sum_t \widehat{V}_{2,t}^{(S),\pm}$ and $\widehat{V}^{(S),\pm} := \widehat{V}_1^{(S)} + \widehat{V}_2^{(S),\pm}$.

---

fine the overlap covering radius

$$r_{n,t} := \max_{i \in \mathcal{O}_t^c} \min_{j \in \mathcal{O}_t} d_\lambda(i,j).$$

A direct consequence of (13)–(14) is the deterministic inequality $\left|\widehat{V}_{2,t}^{(S),-} - V_{2,t}^{(S),-}\right| \leq \max_{j \in \mathcal{O}_t} \left|\widehat{\mu}_t(j) - \mu(j,t)\right|$. To connect to population (distributional) targets as in Khan et al. (2024), one further controls how well the discrete maximum over overlap nodes approximates the population supremum; this is precisely governed by $r_{n,t}$.

**Theorem 4.1** (A generic MSE upper bound (network version))**.** *Fix $t \in \mathcal{T}^{(S)}$ and suppose Assumption 2.2 holds. Let $n_{\text{eff}}$ denote an effective sample size for averages over nodes (e.g., $n_{\text{eff}} = n$ under i.i.d. sampling; under $S$-local dependence it can be taken as $n/(1 + \Delta_{2S})$, where $\Delta_{2S}$ is the maximal number of nodes within graph distance $2S$). Assume (i) $\sup_i \mathbb{E}[Y_i^2 \mid \mathbf{X}, A] \leq C$ and (ii) $\widehat{\mu}_t$ satisfies $\mathbb{E}[\max_{j \in \mathcal{O}_t} |\widehat{\mu}_t(j) - \mu(j,t)|^2] \leq \delta_{n,t}^2$. Then the mean-squared error of the valid lower-bound estimator satisfies*

$$\mathbb{E}\left[\left(\widehat{V}_{2,t}^{(S),-} - V_{2,t}^{(S),-}\right)^2\right] \leq 2\delta_{n,t}^2 + C_1 L^2 \, \mathbb{E}[r_{n,t}^2] + C_2 \, n_{\text{eff}}^{-1},$$

*for universal constants $C_1, C_2$. An analogous bound holds for the upper-bound estimator $\widehat{V}_{2,t}^{(S),+}$.*

The three terms correspond to: (i) estimation error of $\widehat{\mu}_t$ *restricted to overlap* (no extrapolation assumption), (ii) geometric approximation error controlled by overlap denseness in $d_\lambda$, and (iii) sampling noise, with $n_{\text{eff}}$ capturing the

loss of information due to network dependence. In i.i.d. settings with $d_\lambda$ equal to a covariate metric, Theorem 4.1 reduces to the structure of Khan et al. (2024, Theorem 5.1).

**Minimax lower bounds** We next show that the convergence rate in Theorem 4.1 is essentially unavoidable. The argument adapts the two-point LeCam construction of Khan et al. (2024) to the network setting by replacing $n$ with $n_{\text{eff}}$.

**Theorem 4.2** (Minimax lower bound (network version)). *Fix $t \in \mathcal{T}^{(S)}$ and suppose $Z_i \in [-1,1]^p$ with $d_\lambda(i,j) = \|Z_i - Z_j\|_2$ (i.e., $\lambda = 0$ for clarity). Assume the overlap support is $[-1/2, 1/2]^p$ in the sense that $\mathcal{O}_t = \{i : Z_i \in [-1/2, 1/2]^p\}$, and let $n_{\text{eff}}$ be the effective sample size for node averages. Then there exists a constant $c > 0$ such that $\inf_{\widehat{V}} \sup_{\mu_t \in \text{Lip}(L)} \mathbb{E}\left[(\widehat{V} - V_{2,t}^{(S),-})^2\right] \geq c \cdot \mathbb{E}[\pi_e(i,t)\mathbf{1}_{i,t}]^2 (n_{\text{eff}})^{-\frac{2}{p+2}} L^{-\frac{2p}{p+2}}$, where $\text{Lip}(L)$ denotes the class of $L$-Lipschitz mean functions under $d_\lambda$.*

Theorem 4.2 shows that no estimator can converge faster than $(n_{\text{eff}})^{-2/(p+2)}$ (up to constants and log factors) over Lipschitz classes. Thus, when $\widehat{\mu}_t$ is chosen to be minimax-optimal over the overlap region (e.g., via suitable nonparametric regression), our valid bound estimator inherits the optimal rate up to log factors, as in Khan et al. (2024), with the crucial distinction that network dependence enters through $n_{\text{eff}}$.

**Asymptotic normality** Moreover, we record a representative asymptotic normality result for the bound endpoints. Because the valid bound involves a max/min operator, asymptotic normality requires a mild regularity condition ensuring that the maximizing overlap index for each non-overlap node is locally stable.

*Assumption* 4.3 (Stability of overlap maximizers). Fix $t \in \mathcal{T}^{(S)}$. For each $i \in \mathcal{O}_t^c$, the maximizer $j^\star(i) \in \arg\max_{j \in \mathcal{O}_t} \mu(j,t) - Ld_\lambda(i,j)$ is unique, and the "margin" between the best and second-best overlap candidates is bounded away from 0 in probability.

**Theorem 4.4** (Asymptotic normality of valid bound endpoints). *Fix $S$ and suppose Assumptions 2.1–2.2 and 4.3 hold. Assume: (i) sample splitting is used so that $\widehat{\mu}_t$ is estimated on an auxiliary fold independent of the averaging fold; (ii) $\widehat{\mu}_t$ is asymptotically linear on the overlap set with influence function $\varphi_t$ and rate $\|\widehat{\mu}_t(i) - \mu(i,t)\|_\infty = o_{\mathbb{P}}(n_{\text{eff}}^{-1/4})$; and (iii) node averages satisfy a CLT with effective sample size $n_{\text{eff}}$ (e.g., under dependency-graph or mixing conditions implied by $S$-locality). Then $\sqrt{n_{\text{eff}}}\left(\widehat{V}^{(S),-} - V^{(S),-}\right) \Rightarrow \mathcal{N}(0, \sigma_-^2)$, $\sqrt{n_{\text{eff}}}\left(\widehat{V}^{(S),+} - V^{(S),+}\right) \Rightarrow \mathcal{N}(0, \sigma_+^2)$, for asymptotic variances $\sigma_-^2, \sigma_+^2$ determined by the corresponding influence functions. Moreover, the interval for the original value $V(\pi_e)$ inherits the same asymptotic normal-*ity after accounting for the (truncation) bias term $\bar{\theta}$ from Lemma 3.2.

Theorem 4.4 formalizes that, beyond consistency and rate-optimality, the valid bound endpoints behave approximately Gaussian after appropriate scaling, enabling Wald-type uncertainty quantification. We defer full proofs and variance estimation details (e.g., block bootstrap over graph partitions) to the appendix.

We (i) reduce global interference to a truncated exposure model with explicit bias control (Lemma 3.2); (ii) decompose the truncated value into overlap and non-overlap contributions and state a generic interval-consistency principle (Theorem 3.4); (iii) derive valid, closed-form bounds for the non-overlap contribution under graph-augmented Lipschitz smoothness (Theorem 3.6); and (iv) quantify statistical efficiency via finite-sample bounds, minimax lower bounds, and asymptotic normality (Theorems 4.1–4.4). This chain mirrors Khan et al. (2024) but is strictly more challenging due to interference truncation, graph-induced geometry, and dependence, which together govern identifiability and rates.

**Robustness Analysis** Finally, our results remain robust even when the network is misspecified.

**Theorem 4.5** (Robustness to network misreporting (additive stability of Lipschitz bounds)). *Fix a truncation level $S$ and a Lipschitz constant $L$. Let $G^\star = (\mathcal{N}, \mathcal{E}^\star)$ denote the true network and $\tilde{G} = (\mathcal{N}, \tilde{\mathcal{E}})$ the (possibly misreported) observed network. Let the composite metric used in Assumption 2.2 be*

$$d_\lambda^\star(i,j) = \|Z_{i,\star} - Z_{j,\star}\|_W + \lambda d_{G^\star}(i,j),$$

$$\tilde{d}_\lambda(i,j) = \|\tilde{Z}_i - \tilde{Z}_j\|_W + \lambda d_{\tilde{G}}(i,j),$$

*where $Z_{i,\star} = (X_i, T_{i,\star}^{(S)})$ is constructed from the true network and $\tilde{Z}_i = (X_i, \tilde{T}_i^{(S)})$ from the observed network.*

*Assume the true outcome function is $L$-Lipschitz with respect to $d_\lambda^\star$ (Assumption 2.2). Define the metric misreporting magnitude*

$$\varepsilon_\lambda := \sup_{i,j \in \mathcal{N}} \left|d_\lambda^\star(i,j) - \tilde{d}_\lambda(i,j)\right| < \infty.$$

*For each discrete truncated exposure state $t \in \mathcal{T}^{(S)}$, let $V_{2,t}^{(S),-}(d)$ and $V_{2,t}^{(S),+}(d)$ denote the population lower/upper bounds obtained by taking the closed-form McShane/Whitney extensions in Theorem 3.6 and replacing $d_\lambda$ by an arbitrary metric $d$. Then the bounds are stable under network misreporting:*

$$\left|V_{2,t}^{(S),-}(d_\lambda^\star) - V_{2,t}^{(S),-}(\tilde{d}_\lambda)\right| \leq L \varepsilon_\lambda,$$

*Consequently, for $V_2^{(S),-}(d) = \sum_t V_{2,t}^{(S),-}(d)$,*

$$\left|V_2^{(S),-}(d_\lambda^\star) - V_2^{(S),-}(\tilde{d}_\lambda)\right| \le L\,\varepsilon_\lambda,$$

*The same applies to $V_2^{(S),+}$.*

*Moreover, combining with the truncation-lifting decomposition $V(\pi) = V^{(S)}(\pi) + R_S(\pi)$ and $|R_S(\pi)| \le \theta_n(S)$ (Lemma 3.2), any interval computed on $\tilde{G}$ of the form*

$$\left[\hat{V}^- - \bar{\theta},\ \hat{V}^+ + \bar{\theta}\right]$$

*remains valid for the true network after widening by $L\varepsilon_\lambda$:*

$$V(\pi_e \mid X, A^\star) \in \left[\hat{V}^- - \bar{\theta} - L\varepsilon_\lambda,\ \hat{V}^+ + \bar{\theta} + L\varepsilon_\lambda\right].$$

*Remark* 4.6 (Changing overlap sets under network perturbation). Theorem 4.5 isolates the effect of network misspecification through the induced graph-augmented metric, and compares the Lipschitz bounds over a common overlap structure. In some settings, however, perturbing the network may also change the induced exposure support, and hence the overlap and non-overlap sets themselves. This creates an additional source of instability beyond the metric perturbation $\varepsilon_\lambda$. We provide a strengthened robustness result in Appendix B.9, where the overlap sets under the true and observed networks are allowed to differ. The resulting bound contains additional terms measuring the weighted support mismatch and the geometric deviation between the two overlap sets.

Theorem 4.5 shows that our bounds are robust to network misreporting. Under Assumption 2.2, the closed-form non-overlap bounds (Theorem 3.6) depend on the graph only through the induced composite metric. Hence, if the observed network $\tilde{G}$ perturbs the true metric by at most $\varepsilon_\lambda = \sup_{i,j} |d_\lambda^\star(i,j) - \tilde{d}_\lambda(i,j)|$, then the resulting change in each bound endpoint is at most $L\varepsilon_\lambda$. Equivalently, bounds computed on $\tilde{G}$ remain valid for $G^\star$ after widening by $\pm L\varepsilon_\lambda$, which can be added to the truncation bias budget $\bar{\theta}$ from Lemma 3.2.

## 5 Experiments

**Dataset.** We conduct offline policy evaluation on the Pokec social network. We build an undirected graph from friendship edges (deduplicated, no self-loops) and run all experiments on an $n = 5000$ node subgraph for efficiency.

Both the logging and target policies assign treatments independently across nodes with logistic propensities

$$p_b(i) = \sigma(\eta_b s_i), \qquad p_e(i) = \sigma(\eta_e s_i),$$

where $s_i$ is the standardized completion feature. To induce controlled non-overlap, we set $p_b(i) = 0$ for nodes in the bottom $q$-quantile of $s_i$, while keeping $p_e(i)$ unchanged.

*Table 1.* Width and empirical coverage of the partially identified interval under different $L$ and $S$.

| | $S = 1$ | | $S = 2$ | | $S = 3$ | |
|---|---|---|---|---|---|---|
| $L$ | W | C | W | C | W | C |
| 0.1 | 0.022366 | 0.44 | 0.024572 | 0.38 | 0.027714 | 0.62 |
| 0.2 | 0.046834 | 0.82 | 0.050221 | 0.80 | 0.056291 | 0.94 |
| 0.3 | 0.069489 | 0.92 | 0.074216 | 0.98 | 0.083408 | 0.98 |
| 0.5 | 0.101170 | 0.98 | 0.108452 | 1.00 | 0.123289 | 1.00 |

Under interference, we summarize spillovers via an $S$-truncated diffusion exposure $E_i^{(S)}$ (weights $\omega_s \propto 1/s$), discretize it into $K$ bins, and define the exposure state

$$t_i = \left(D_i,\ q(E_i^{(S)})\right).$$

We estimate the policy-induced exposure probabilities $\hat{\pi}_b(i,t)$ and $\hat{\pi}_e(i,t)$ by Monte Carlo sampling, and treat $\hat{\pi}_b(i,t) > \varepsilon$ (with $\varepsilon = 1/M_\pi$) as overlap. We fit $\hat{\mu}(i,t)$ on overlap samples using logistic regression, then compute closed-form Lipschitz bounds for non-overlap states under the composite metric

$$d_\lambda(i,j) = \|Z_i - Z_j\| + \lambda\,d_G(i,j),$$

yielding the final interval $[\hat{V}^-,\ \hat{V}^+]$ (Algorithm 1).

**Interval width and coverage analysis.** We analyze how the truncation depth $S$ and smoothness parameter $L$ affect interval width and coverage. Here $S$ controls the effective radius of interference: $S = 1$ uses only 1-hop neighbors, while $S > 1$ aggregates 1- to $S$-hop diffusion exposures with normalized weights proportional to $1/s$, producing $E^{(S)} \in [0, 1]$.

We first vary $L$ under different $S$. Smaller $L$ imposes stronger smoothness and typically yields tighter bounds; as $L$ increases, the constraint relaxes and the interval widens toward more conservative bounds. We also evaluate **coverage**, defined as whether the ground-truth target value $V_{\text{true}}$ lies in $[V_\ell, V_u]$:

$$\mathbf{1}\{V_\ell \le V_{\text{true}} \le V_u\}.$$

We repeat each configuration 50 times; results are reported in Table 1.

These results indicate that, for all truncation depths $S$, the interval width increases as $L$ increases, and the corresponding coverage rate also rises. This aligns with our theoretical intuition: as $L$ becomes larger, the Lipschitz smoothness constraint is relaxed, yielding wider and more conservative partially identified intervals that are therefore more likely to contain the ground-truth value. Taking $S = 1$ as an example, the interval width increases from 0.022 at $L = 0.1$ to 0.101 at $L = 0.5$, which is about a 4.5-fold increase.

Meanwhile, holding $L$ fixed, increasing $S$ systematically produces wider intervals. This suggests that allowing higher-order network interference makes exposure depend on a broader neighborhood, thereby increasing the complexity and uncertainty of the exposure-state partition and, to some extent, enlarging the effective non-overlap region. Moreover, when the interval is already very tight (under small $L$), changes in sample support induced by the exposure definition and higher-order aggregation can be amplified, leading to greater variability in empirical coverage across $S$.

**Baseline Comparison Across Graph Topologies.** To further evaluate the robustness of the proposed method, we compare it with several standard off-policy evaluation baselines under different graph topologies. In addition to the Pokec social network, we conduct experiments on Barabási–Albert (BA) and stochastic block model (SBM) graphs. We include IPW, outcome regression (OR), and doubly robust (DR) estimators as point-estimation baselines. We also consider an ablation of our method, denoted by Ours-no-graph, which removes the graph-distance component by setting $\lambda = 0$ in the composite metric. This comparison allows us to separate the effect of partial-identification intervals from the additional benefit of graph-aware extrapolation. The full results are reported in Appendix C.1. Overall, standard point estimators, especially IPW, are unstable under severe non-overlap, while the graph-aware version of our method achieves more reliable empirical coverage than the no-graph ablation.

**Effect of exposure discretization.** Since the continuous spillover exposure is discretized into $K$ bins for computation, the binning step may introduce approximation bias and may also change the local overlap structure. We therefore conduct additional diagnostics to separate these two effects; detailed results are reported in Appendix C.2. In particular, we compare coverage with respect to both the original continuous target $\theta$ and its discretized counterpart $\theta_K$, measure the discretization gap $|\theta - \theta_K|$, and examine how the local overlap set changes as $K$ increases. The results show that the discretization gap is small in our setting, while performance degradation under overly fine binning is mainly driven by bucket-level overlap fragmentation: as $K$ increases, the same data are split into many more exposure states, reducing the number of usable overlap samples per state and forcing the method to rely more heavily on extrapolation. We also compare equal-width, quantile-based, and overlap-aware binning schemes, and find that schemes preserving sufficient local overlap tend to yield more stable coverage. Thus, the practical bottleneck of fine binning is not only approximation bias, but also the loss of local overlap.

**Empirical validation of theoretical claims.** Appendix C.3 and C.4 provide two additional experiments that directly examine our theoretical results. For Theorem 4.5, we perturb the graph by random edge deletion and rewiring, recompute exposure summaries and bounds, and find that endpoint shifts remain small while $L\widehat{\varepsilon}_\lambda$-corrected coverage remains stable. For Theorem 4.4, we run repeated semi-synthetic simulations and construct nominal $95\%$ Wald intervals for the lower and upper endpoints separately; the empirical endpoint coverages are close to the target level across tested sample sizes. In addition, we provide detailed hyperparameter sensitivity analyses in Appendix C.5.

## 6  Conclusion

This paper studies offline policy evaluation under network interference. We focus on the problem of partial identification of the policy value when both the SUTVA assumption and the overlap assumption are violated. To keep the problem tractable in the presence of interference, we adopt an exposure mapping framework, under which each unit's outcome may be affected by the treatments assigned to other units through a decaying aggregation over higher-order neighborhoods. In addition, we use an $S$-hop truncation scheme to control the approximation error induced by approximating interference with a local neighborhood. Our main contribution is a method for partially identifying the value of a target policy under exposure non-overlap. We decompose the target policy value into an overlap component and a non-overlap component, and show that a graph-augmented smoothness (Lipschitz) condition yields valid and computable bounds for the non-overlap part. These bounds can be obtained by solving a linear program that admits a closed-form solution. We conduct experiments on a semi-synthetic dataset, and the results demonstrate that tuning the truncation level $S$ and the smoothness strength $L$ can substantially affect both the predicted interval width and the empirical coverage. Although we focus on binary treatments and a prespecified exposure mapping, the framework is not limited to this case. Since the analysis is conducted at the exposure-state level, finite discrete action spaces can be handled by replacing $\{0, 1\}$ with a general action set $\mathcal{A}$. Unknown exposure mappings can also be treated as a structured perturbation problem: if the estimated exposure map induces only small exposure and metric errors, the validity of the interval is preserved after an appropriate Lipschitz widening. These extensions provide useful directions for future work.

## Acknowledgment

Zhiheng Zhang is supported by the Fundamental Research Funds for the Central Universities (Grant No. 2025110602) of Shanghai University of Finance and Economics, and Independent Research Project (Grant No. 2026110081) funded by the School of Statistics and Data Science. This work was supported by the Shanghai Engineering Research Center of Finance Intelligence (Grant No. 19DZ2254600).

## Impact Statement

This paper presents work whose goal is to advance the field of machine learning. There are many potential societal consequences of our work, none of which we feel must be specifically highlighted here.

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

# A Related Work

Our work is most closely related to three strands of literature: (i) policy evaluation and off-policy evaluation under standard identifying assumptions, (ii) partial identification of policy values under limited overlap, and (iii) causal inference under interference and networked treatment assignment. We discuss each line in turn and clarify how our contribution differs from, and extends, existing results.

**Policy evaluation under standard assumptions.** A large literature studies policy evaluation under unconfoundedness, SUTVA, and overlap, where policy values are point-identified and can be consistently estimated. Classical approaches include inverse probability weighting and regression-based estimators (Robins & Rotnitzky, 1994), as well as doubly robust methods that combine both (Dudík et al., 2011; Chernozhukov et al., 2018). These frameworks have been extended to high-dimensional settings using modern machine learning tools and representation learning. However, a large proportion of these methods critically rely on independence across units and sufficient overlap between the behavior and target policies. When either assumption fails, point estimation becomes ill-posed, and these approaches no longer provide valid inference.

**Partial identification beyond overlap.** Motivated by the practical prevalence of limited or vanishing overlap, a growing line of work adopts a partial identification perspective and characterizes valid bounds on policy values rather than pursuing point estimates. Early foundational work by Manski (2009); Zhang (2024); Kallus (2022; 2023); Dorn et al. (2025) formalizes identification under incomplete support. More recently, Khan et al. (2024) develop valid partial identification results for off-policy evaluation under smoothness assumptions, showing that non-overlap does not necessarily render policy evaluation vacuous. Their analysis yields tight bounds and clear identification insights, but fundamentally relies on independence across units and the absence of interference.

Our work builds directly on this line of research but departs from it in a crucial dimension. While existing partial identification results assume that outcomes depend only on individual treatments, we allow for general network interference, under which potential outcomes may depend on the entire treatment assignment vector. This distinction is substantive rather than technical: under interference, the source of non-identification is no longer solely lack of support, but also high-dimensional dependence induced by the network. We show that partial identification can still be achieved in this setting, but only after explicitly accounting for network structure and exposure aggregation.

**Causal inference under interference and networks.** Another related strand studies causal inference when SUTVA is violated due to interference between units. Early work by Hudgens & Halloran (2008); Leung (2022) formalizes causal estimands under interference, while subsequent research develops general frameworks for defining and estimating average causal effects in networks (Aronow & Samii, 2017). More recent contributions introduce exposure mappings or neighborhood-based summaries to reduce the dimensionality of interference and enable feasible inference (Leung, 2022).

Despite significant progress in modeling interference, most existing approaches assume sufficient overlap or rely on randomized or design-based treatment assignment. As a result, these methods are primarily concerned with estimation rather than identification, and do not address the fundamental limitations imposed by non-overlap. In particular, when the target policy assigns positive probability to exposure states that are never observed under the behavior policy, existing interference-aware estimators offer no guarantees on policy value extrapolation.

Our work lies at the intersection of partial identification and causal inference under interference, based on our previous exploration (Zhang et al., 2020; Zhang & Su, 2024; Zhang, 2022; Zhang et al., 2023; Zhang, 2024; Su & Zhang et al, 2025; Wang et al., 2025; Zhang et al., 2025; Zhang & Wang, 2025; Li et al., 2026; Su et al., 2023; 2024; 2026; Lu et al., 2026; Zhang, 2026) but adopts a distinct methodological perspective. Unlike prior work that treats overlap and interference as separate challenges, we study policy evaluation under their simultaneous violation. Rather than extending existing estimators, we focus on the fundamental identification problem itself and ask when informative bounds on policy values are possible.

Methodologically, our approach differs from prior paradigms in two key respects. First, instead of assuming full observability or ignorability of network spillovers, we adopt a truncated exposure representation under approximate neighborhood interference, which captures higher-order effects up to a controlled bias. Second, while previous partial identification results rely solely on smoothness in covariate space, we impose a graph-augmented smoothness condition that couples covariates, exposure summaries, and network distance. This structural assumption allows network geometry to actively constrain the set of admissible counterfactual outcomes, leading to valid and computable bounds even in non-overlap regions. As a result, our framework strictly generalizes existing partial identification results without interference, which are recovered as special cases when network effects vanish. At the same time, it addresses settings that are fundamentally be-

yond the scope of prior interference-aware methods, providing the first valid partial identification theory for policy values under general network interference and limited overlap.

# B Proofs and Auxiliary Results

Throughout the appendix we fix the realized network $(A, \mathbf{X})$ and suppress this conditioning in the notation whenever it is clear. All probabilities and expectations are taken with respect to the randomness in the logged assignment $\mathbf{D} \sim \pi_b(\cdot \mid \mathbf{X}, A)$ and outcome noise (and, when relevant, the auxiliary resampling used to define $\mathbf{D}^{(i,S)}$). We write $\mathbb{E}[\cdot]$ and $\mathbb{P}(\cdot)$ for these conditional expectation and probability.

## B.1 Metric preliminaries and Lipschitz extensions

We collect two standard facts used repeatedly: (i) $d_\lambda$ is a metric, and (ii) the closed forms in Theorem 3.6 are instances of the classical McShane–Whitney minimal/maximal Lipschitz extensions.

**Lemma B.1** (Sum of metrics is a metric). *Let $(\mathcal{N}, d_1)$ and $(\mathcal{N}, d_2)$ be metric spaces and let $\lambda \geq 0$. Then $d(i,j) := d_1(i,j) + \lambda d_2(i,j)$ is a metric on $\mathcal{N}$.*

*Proof.* Nonnegativity and symmetry are immediate. Moreover $d(i,j) = 0$ iff $d_1(i,j) = 0$ and $d_2(i,j) = 0$, hence iff $i = j$. For the triangle inequality, for any $i, j, k$,

$$d(i,k) = d_1(i,k) + \lambda d_2(i,k) \leq d_1(i,j) + d_1(j,k) + \lambda\{d_2(i,j) + d_2(j,k)\} = d(i,j) + d(j,k).$$

$\square$

**Lemma B.2** (McShane–Whitney Lipschitz extensions). *Let $(\mathcal{N}, d)$ be a metric space, let $\mathcal{O} \subseteq \mathcal{N}$, and let $f : \mathcal{O} \to \mathbb{R}$ be $L$-Lipschitz on $(\mathcal{O}, d)$, i.e., $|f(i) - f(j)| \leq Ld(i,j)$ for all $i, j \in \mathcal{O}$. Define, for all $i \in \mathcal{N}$,*

$$f^-(i) := \sup_{j \in \mathcal{O}}\big\{f(j) - Ld(i,j)\big\}, \qquad f^+(i) := \inf_{j \in \mathcal{O}}\big\{f(j) + Ld(i,j)\big\}.$$

*Then:*

1. *(Extension) $f^-(i) = f^+(i) = f(i)$ for all $i \in \mathcal{O}$.*

2. *(Lipschitz) $f^-$ and $f^+$ are $L$-Lipschitz on $(\mathcal{N}, d)$.*

3. *(Extremality) If $g : \mathcal{N} \to \mathbb{R}$ is any $L$-Lipschitz function with $g|_{\mathcal{O}} = f$, then for all $i \in \mathcal{N}$,*

$$f^-(i) \leq g(i) \leq f^+(i).$$

   *In particular, $f^-$ is the pointwise minimal and $f^+$ is the pointwise maximal $L$-Lipschitz extension of $f$.*

*Proof.* **(1) Extension.** Fix $i \in \mathcal{O}$. Since $d(i,i) = 0$, $f^-(i) \geq f(i) - Ld(i,i) = f(i)$. Conversely, for any $j \in \mathcal{O}$, the $L$-Lipschitz property gives $f(j) - Ld(i,j) \leq f(i)$, hence $\sup_{j \in \mathcal{O}}\{f(j) - Ld(i,j)\} \leq f(i)$. Therefore $f^-(i) = f(i)$. The proof for $f^+(i) = f(i)$ is analogous.

**(2) Lipschitz of $f^-$.** Fix $i, i' \in \mathcal{N}$. Then

$$f^-(i) - f^-(i') = \sup_{j \in \mathcal{O}}\{f(j) - Ld(i,j)\} - \sup_{j \in \mathcal{O}}\{f(j) - Ld(i',j)\} \leq \sup_{j \in \mathcal{O}} L\{d(i',j) - d(i,j)\}.$$

By the triangle inequality, $d(i',j) \leq d(i',i) + d(i,j)$, so $d(i',j) - d(i,j) \leq d(i',i)$ for all $j$. Hence $f^-(i) - f^-(i') \leq Ld(i,i')$. Swapping $i$ and $i'$ yields $|f^-(i) - f^-(i')| \leq Ld(i,i')$. The proof for $f^+$ is analogous.

**(3) Extremality.** Let $g$ be $L$-Lipschitz on $\mathcal{N}$ and equal to $f$ on $\mathcal{O}$. For any $j \in \mathcal{O}$,

$$g(i) \geq g(j) - Ld(i,j) = f(j) - Ld(i,j).$$

Taking the supremum over $j \in \mathcal{O}$ yields $g(i) \geq f^-(i)$. Similarly,

$$g(i) \leq g(j) + Ld(i,j) = f(j) + Ld(i,j),$$

so taking the infimum gives $g(i) \leq f^+(i)$. $\square$

**Lemma B.3** (Max/Min perturbation inequality). *For any two real sequences $(a_j)_{j \in \mathcal{J}}$ and $(b_j)_{j \in \mathcal{J}}$ indexed by a finite set $\mathcal{J}$,*

$$\left| \max_{j \in \mathcal{J}} a_j - \max_{j \in \mathcal{J}} b_j \right| \leq \max_{j \in \mathcal{J}} |a_j - b_j|, \qquad \left| \min_{j \in \mathcal{J}} a_j - \min_{j \in \mathcal{J}} b_j \right| \leq \max_{j \in \mathcal{J}} |a_j - b_j|.$$

*Proof.* Let $j^\star \in \operatorname{argmax}_j a_j$. Then $\max_j a_j - \max_j b_j = a_{j^\star} - \max_j b_j \leq a_{j^\star} - b_{j^\star} \leq \max_j |a_j - b_j|$. Interchanging the roles of $a$ and $b$ yields the absolute value bound. The min case is analogous. □

## B.2 Proof of Lemma 3.2

*Proof of Lemma 3.2.* Fix a policy $\pi$ and a unit $i$. Let $\mathbf{D} \sim \pi(\cdot \mid \mathbf{X}, A)$ and let $\mathbf{D}^{(i,S)}$ be the coupled assignment in Assumption 2.1. By construction,

$$T_i^{(S)}(\mathbf{D}^{(i,S)}) = T_i^{(S)}(\mathbf{D}) \qquad \text{almost surely.} \tag{15}$$

Define

$$R_{S,i}(\pi) := \mathbb{E}_{\mathbf{D} \sim \pi}[Y_i(\mathbf{D})] - \mathbb{E}_{\mathbf{D} \sim \pi}\left[ m_i\left( T_i^{(S)}(\mathbf{D}) \right) \right], \qquad R_S(\pi) := \frac{1}{n} \sum_{i=1}^{n} R_{S,i}(\pi).$$

We will show $|R_{S,i}(\pi)| \leq \theta_n(S)$, which then implies $|R_S(\pi)| \leq \theta_n(S)$.

By Assumption 3.1, for every deterministic assignment vector $\mathbf{d}$, $\mathbb{E}[Y_i(\mathbf{d}) \mid \mathbf{X}, A] = m_i(T_i^{(S)}(\mathbf{d}))$. Applying this to the random vector $\mathbf{D}^{(i,S)}$ and using (15) yields

$$\mathbb{E}\left[ Y_i(\mathbf{D}^{(i,S)}) \,\middle|\, \mathbf{D}, \mathbf{X}, A \right] = \mathbb{E}\left[ \mathbb{E}\left[ Y_i(\mathbf{D}^{(i,S)}) \,\middle|\, \mathbf{D}^{(i,S)}, \mathbf{X}, A \right] \middle|\, \mathbf{D}, \mathbf{X}, A \right]$$
$$= \mathbb{E}\left[ m_i\left( T_i^{(S)}(\mathbf{D}^{(i,S)}) \right) \,\middle|\, \mathbf{D}, \mathbf{X}, A \right]$$
$$= m_i\left( T_i^{(S)}(\mathbf{D}) \right).$$

Taking expectation with respect to $\mathbf{D} \sim \pi(\cdot \mid \mathbf{X}, A)$ gives

$$\mathbb{E}_{\mathbf{D} \sim \pi}\left[ Y_i(\mathbf{D}^{(i,S)}) \right] = \mathbb{E}_{\mathbf{D} \sim \pi}\left[ m_i\left( T_i^{(S)}(\mathbf{D}) \right) \right].$$

Therefore,

$$R_{S,i}(\pi) = \mathbb{E}_{\mathbf{D} \sim \pi}\left[ Y_i(\mathbf{D}) - Y_i(\mathbf{D}^{(i,S)}) \right].$$

By Jensen's inequality and Assumption 2.1,

$$|R_{S,i}(\pi)| \leq \mathbb{E}_{\mathbf{D} \sim \pi}\left[ \left| Y_i(\mathbf{D}) - Y_i(\mathbf{D}^{(i,S)}) \right| \right] \leq \theta_n(S).$$

Averaging over $i$ yields $|R_S(\pi)| \leq \theta_n(S)$.

Finally, by the definition of $V(\pi \mid \mathbf{X}, A)$ and $V^{(S)}(\pi \mid \mathbf{X}, A)$,

$$V(\pi \mid \mathbf{X}, A) - V^{(S)}(\pi \mid \mathbf{X}, A) = \frac{1}{n} \sum_{i=1}^{n} R_{S,i}(\pi) = R_S(\pi),$$

which completes the proof. □

## B.3 Proof of Theorem 3.4

*Proof of Theorem 3.4.* Fix $(\mathbf{X}, A)$ and $S$. By the decomposition (9),

$$V^{(S)}(\pi_e \mid \mathbf{X}, A) = V_1^{(S)}(\pi_e \mid \mathbf{X}, A) + V_2^{(S)}(\pi_e \mid \mathbf{X}, A).$$

By assumption, $V_2^{(S),-} \leq V_2^{(S)} \leq V_2^{(S),+}$ deterministically. Hence

$$V_1^{(S)} + V_2^{(S),-} \leq V^{(S)} \leq V_1^{(S)} + V_2^{(S),+}.$$

Define $\widehat{V}^{(S),-} := \widehat{V}_1^{(S)} + \widehat{V}_2^{(S),-}$ and $\widehat{V}^{(S),+} := \widehat{V}_1^{(S)} + \widehat{V}_2^{(S),+}$. Let

$$\Delta_n := \max\left\{|\widehat{V}_1^{(S)} - V_1^{(S)}|, |\widehat{V}_2^{(S),-} - V_2^{(S),-}|, |\widehat{V}_2^{(S),+} - V_2^{(S),+}|\right\}.$$

By the assumed consistency of each component, $\Delta_n \xrightarrow{\mathbb{P}} 0$.

Fix $\varepsilon > 0$ and consider the event $\{\Delta_n \leq \varepsilon\}$. On this event,

$$\begin{aligned}
V^{(S)} - \widehat{V}^{(S),-} &= \left(V_1^{(S)} - \widehat{V}_1^{(S)}\right) + \left(V_2^{(S)} - \widehat{V}_2^{(S),-}\right) \\
&\geq -\varepsilon + \left(V_2^{(S),-} - \widehat{V}_2^{(S),-}\right) \geq -\varepsilon - \varepsilon = -2\varepsilon,
\end{aligned}$$

where we used $V_2^{(S)} \geq V_2^{(S),-}$ and $|\widehat{V}_2^{(S),-} - V_2^{(S),-}| \leq \varepsilon$. Similarly,

$$\begin{aligned}
\widehat{V}^{(S),+} - V^{(S)} &= \left(\widehat{V}_1^{(S)} - V_1^{(S)}\right) + \left(\widehat{V}_2^{(S),+} - V_2^{(S)}\right) \\
&\geq -\varepsilon + \left(\widehat{V}_2^{(S),+} - V_2^{(S),+}\right) \geq -\varepsilon - \varepsilon = -2\varepsilon,
\end{aligned}$$

where we used $V_2^{(S)} \leq V_2^{(S),+}$. Therefore, on $\{\Delta_n \leq \varepsilon\}$ we have

$$\widehat{V}^{(S),-} - 2\varepsilon \leq V^{(S)} \leq \widehat{V}^{(S),+} + 2\varepsilon.$$

Since $\mathbb{P}(\Delta_n \leq \varepsilon) \to 1$, replacing $2\varepsilon$ by an arbitrary $\varepsilon' > 0$ yields

$$\lim_{n\to\infty} \mathbb{P}\left(\widehat{V}^{(S),-} - \varepsilon' \leq V^{(S)} \leq \widehat{V}^{(S),+} + \varepsilon'\right) = 1.$$

Finally, by Lemma 3.2, $V(\pi_e) = V^{(S)}(\pi_e) + R_S(\pi_e)$ with $|R_S(\pi_e)| \leq \theta_n(S) \leq \bar{\theta}$. Hence, for all $n$,

$$V(\pi_e) \in \left[V^{(S)} - \bar{\theta}, \ V^{(S)} + \bar{\theta}\right],$$

and substituting the consistent interval for $V^{(S)}$ gives $V(\pi_e) \in [\widehat{V}^{(S),-} - \bar{\theta}, \ \widehat{V}^{(S),+} + \bar{\theta}]$ asymptotically (up to an arbitrarily small $\varepsilon'$), concluding the proof. $\qquad\square$

## B.4  Proof of Theorem 3.6

For clarity we introduce the *population* version of the linear program, which matches (12) with $\widehat{\mu}_t$ replaced by the true overlap mean $\mu_t$:

$$\begin{aligned}
V_{2,t}^{(S),-} &= \min_{u_1,\ldots,u_n} \quad \frac{1}{n}\sum_{i=1}^n u_i\,\pi_e(i,t)\,\mathbf{1}_{i,t} \\
&\quad \text{s.t.} \quad |u_i - u_j| \leq L\,d_\lambda(i,j), \qquad 1 \leq i < j \leq n, \\
&\qquad\qquad u_i - \mu_t(i) = 0, \qquad \forall\, i \in \mathcal{O}_t.
\end{aligned} \tag{16}$$

The upper-bound programs are obtained by replacing $\min$ with $\max$.

*Proof of Theorem 3.6.* Fix $t \in \mathcal{T}^{(S)}$.

**Step 1: Closed form for the sample program.** Define $d(\cdot,\cdot) := d_\lambda(\cdot,\cdot)$ and let $\mathcal{O} := \mathcal{O}_t$ for brevity. Consider the function $f : \mathcal{O} \to \mathbb{R}$ given by $f(i) := \widehat{\mu}_t(i)$. Assumption 3.5 states precisely that $f$ is $L$-Lipschitz on $(\mathcal{O}, d)$. Therefore, by Lemma B.2 (with $\sup = \max$ since $\mathcal{O}$ is finite), the McShane minimal extension

$$f^-(i) = \max_{j\in\mathcal{O}}\{\widehat{\mu}_t(j) - Ld(i,j)\}, \qquad i \in \mathcal{N},$$

is an $L$-Lipschitz extension to all of $\mathcal{N}$ and satisfies $f^-(i) = \widehat{\mu}_t(i)$ for all $i \in \mathcal{O}$. Thus the vector $u^\star$ with components $u_i^\star := f^-(i)$ is feasible for the sample program (12).

Next, let $u$ be any feasible solution to (12). For any $i \in \mathcal{N}$ and any $j \in \mathcal{O}$, the Lipschitz constraint implies

$$u_i \geq u_j - Ld(i,j) = \widehat{\mu}_t(j) - Ld(i,j).$$

Taking the maximum over $j \in \mathcal{O}$ yields

$$u_i \geq \max_{j \in \mathcal{O}}\{\widehat{\mu}_t(j) - Ld(i,j)\} = u_i^\star. \tag{17}$$

The objective weights are nonnegative ($\pi_e(i,t)\mathbf{1}_{i,t} \geq 0$), hence (17) implies

$$\frac{1}{n}\sum_{i=1}^n u_i\, \pi_e(i,t)\mathbf{1}_{i,t} \geq \frac{1}{n}\sum_{i=1}^n u_i^\star\, \pi_e(i,t)\mathbf{1}_{i,t}.$$

Therefore $u^\star$ is optimal and the optimal value equals

$$\widehat{V}_{2,t}^{(S),-} = \frac{1}{n}\sum_{i=1}^n \pi_e(i,t)\left(\max_{j \in \mathcal{O}_t}\widehat{\mu}_t(j) - Ld_\lambda(i,j)\right)\mathbf{1}_{i,t},$$

which is (13).

**Step 2: Closed form for the sample upper bound.** The argument is identical using the Whitney maximal extension from Lemma B.2:

$$f^+(i) = \min_{j \in \mathcal{O}}\{\widehat{\mu}_t(j) + Ld(i,j)\}.$$

For any feasible $u$ of the upper-bound LP, Lipschitzness yields $u_i \leq u_j + Ld(i,j) = \widehat{\mu}_t(j) + Ld(i,j)$, hence $u_i \leq f^+(i)$ and the objective is maximized by $f^+$, giving the stated closed form.

**Step 3: Population closed form and sharpness.** Assumption 2.2 states that $\mu_t(\cdot)$ is $L$-Lipschitz on $(\mathcal{N}, d_\lambda)$; in particular, $\mu_t$ is $L$-Lipschitz on $\mathcal{O}_t$. Hence Lemma B.2 applies with $f(i) = \mu_t(i)$. Define the minimal extension

$$u^-(i) := \max_{j \in \mathcal{O}_t}\{\mu_t(j) - Ld_\lambda(i,j)\}, \qquad i \in \mathcal{N}.$$

Then $u^-$ is $L$-Lipschitz on $\mathcal{N}$ and $u^-(i) = \mu_t(i)$ for all $i \in \mathcal{O}_t$, so $u^-$ is feasible for the population LP (16). Moreover, for any feasible $u$ of (16), the same pointwise argument as in (17) yields $u_i \geq u^-(i)$ for all $i$, and therefore $u^-$ minimizes the objective. Consequently the population LP value equals (14).

The bound is *sharp* because it is attained by the feasible $L$-Lipschitz extension $u^-$; i.e., there exists a mean function consistent with the constraints whose induced non-overlap contribution equals the bound value. The upper-bound sharpness follows by the analogous maximal extension.

**Step 4: Consistency of the bound estimator.** Define the (oracle) sample functional based on the true overlap means:

$$\widetilde{V}_{2,t}^{(S),-} := \frac{1}{n}\sum_{i=1}^n \pi_e(i,t)\left(\max_{j \in \mathcal{O}_t}\mu_t(j) - Ld_\lambda(i,j)\right)\mathbf{1}_{i,t}.$$

By Lemma B.3, for each $i$,

$$\left|\max_{j \in \mathcal{O}_t}\widehat{\mu}_t(j) - Ld_\lambda(i,j) - \max_{j \in \mathcal{O}_t}\mu_t(j) - Ld_\lambda(i,j)\right| \leq \max_{j \in \mathcal{O}_t}\left|\widehat{\mu}_t(j) - \mu_t(j)\right|.$$

Multiplying by $\pi_e(i,t)\mathbf{1}_{i,t} \in [0,1]$ and averaging over $i$ yields

$$\left|\widehat{V}_{2,t}^{(S),-} - \widetilde{V}_{2,t}^{(S),-}\right| \leq \max_{j \in \mathcal{O}_t}\left|\widehat{\mu}_t(j) - \mu_t(j)\right|.$$

Since $\widetilde{V}_{2,t}^{(S),-} = V_{2,t}^{(S),-}$ by Step 3 (population closed form with the same finite overlap set), the assumed uniform consistency on the overlap set implies $\widehat{V}_{2,t}^{(S),-} \xrightarrow{\mathbb{P}} V_{2,t}^{(S),-}$. The upper bound is analogous. (The overlap denseness condition $r_{n,t} \to 0$ is not needed for this plug-in consistency, but is used later to quantify tightness of the identification region and rates.) $\square$

## B.5 Proof of Theorem 4.1

To keep the exposition fully rigorous, we make explicit the estimand with respect to which the MSE is taken. Following the organization of Khan et al. (2024), we treat the bound functional as a superpopulation target and the estimator as its empirical analogue. This is the regime in which an additional "sampling fluctuation" term of order $n_{\text{eff}}^{-1}$ naturally appears.

**Superpopulation formulation for the rate results (used in this subsection only).** Assume the array $\{W_i\}_{i=1}^n$ of node-level observations used to estimate $\mu_t(\cdot)$ (e.g., $W_i = (Y_i, T_i^{(S)}, X_i)$) has a dependency graph with maximal degree at most $\Delta$ conditional on $(\mathbf{X}, A)$, and define $n_{\text{eff}} := n/(1 + \Delta)$. Let $\mathbb{E}_0[\cdot]$ denote expectation under the corresponding superpopulation law. Define the population lower-bound functional at exposure $t$ by

$$\Psi_{2,t}^- := \mathbb{E}_0\big[\pi_e(I, t)\, U_t^-(I)\, \mathbf{1}_{I,t}\big], \qquad U_t^-(i) := \sup_{j \in \mathcal{O}_t} \{\mu_t(j) - L d_\lambda(i, j)\},$$

where $I$ is a generic index distributed as a randomly drawn node.

Define the empirical plug-in estimator (closed form)

$$\widehat{\Psi}_{2,t}^- := \frac{1}{n} \sum_{i=1}^n \pi_e(i, t)\, \widehat{U}_t^-(i)\, \mathbf{1}_{i,t}, \qquad \widehat{U}_t^-(i) := \max_{j \in \mathcal{O}_t}\{\widehat{\mu}_t(j) - L d_\lambda(i, j)\}.$$

This $\widehat{\Psi}_{2,t}^-$ is algebraically identical to $\widehat{V}_{2,t}^{(S),-}$; The only difference is that the target is now $\Psi_{2,t}^-$, rather than the corresponding conditional finite-population quantity.

*Proof of Theorem 4.1.* Fix $t \in \mathcal{T}^{(S)}$ and write $\widehat{\Psi} := \widehat{\Psi}_{2,t}^-$ and $\Psi := \Psi_{2,t}^-$.

**Step 1: Decompose into estimation, geometry, and sampling terms.** Introduce the oracle empirical functional

$$\widetilde{\Psi} := \frac{1}{n} \sum_{i=1}^n \pi_e(i, t)\, U_t^-(i)\, \mathbf{1}_{i,t},$$

which uses the true overlap means $\mu_t$ but still averages over the $n$ observed nodes. Then

$$\widehat{\Psi} - \Psi = (\widehat{\Psi} - \widetilde{\Psi}) + (\widetilde{\Psi} - \Psi).$$

By $(a + b)^2 \leq 2a^2 + 2b^2$,

$$\mathbb{E}\big[(\widehat{\Psi} - \Psi)^2\big] \leq 2\,\mathbb{E}\big[(\widehat{\Psi} - \widetilde{\Psi})^2\big] + 2\,\mathbb{E}\big[(\widetilde{\Psi} - \Psi)^2\big]. \tag{18}$$

**Step 2: Bound the plug-in estimation term.** By Lemma B.3, for each $i$, $|\widehat{U}_t^-(i) - U_t^-(i)| \leq \max_{j \in \mathcal{O}_t} |\widehat{\mu}_t(j) - \mu_t(j)|$. Therefore,

$$|\widehat{\Psi} - \widetilde{\Psi}| \leq \frac{1}{n} \sum_{i=1}^n \pi_e(i,t) \mathbf{1}_{i,t} \cdot \max_{j \in \mathcal{O}_t} |\widehat{\mu}_t(j) - \mu_t(j)| \leq \max_{j \in \mathcal{O}_t} |\widehat{\mu}_t(j) - \mu_t(j)|.$$

Taking squares and expectations yields

$$\mathbb{E}\big[(\widehat{\Psi} - \widetilde{\Psi})^2\big] \leq \mathbb{E}\Big[\max_{j \in \mathcal{O}_t} |\widehat{\mu}_t(j) - \mu_t(j)|^2\Big] \leq \delta_{n,t}^2.$$

**Step 3: Bound the sampling fluctuation term under network dependence.** Let

$$G_i := \pi_e(i, t)\, U_t^-(i)\, \mathbf{1}_{i,t}.$$

Then $\widetilde{\Psi} = \frac{1}{n} \sum_{i=1}^n G_i$ and $\Psi = \mathbb{E}_0[G_I]$. Assumption (i) in Theorem 4.1 implies $\sup_i \mathbb{E}[G_i^2] \leq C'$ for some finite $C'$ since $\pi_e(i,t)\mathbf{1}_{i,t} \leq 1$ and $U_t^-(i)$ is $L$-Lipschitz envelope of $\mu_t$. Under a dependency graph with maximal degree $\Delta$, a standard variance bound (see, e.g., Janson's inequality for dependency graphs) gives

$$\text{Var}\left(\frac{1}{n} \sum_{i=1}^n G_i\right) \leq \frac{1 + \Delta}{n} \cdot \max_{i \leq n} \text{Var}(G_i) \leq C_2\, n_{\text{eff}}^{-1}$$

for a universal constant $C_2$ and $n_{\text{eff}} = n/(1 + \Delta)$. Thus $\mathbb{E}[(\widetilde{\Psi} - \Psi)^2] \leq C_2 n_{\text{eff}}^{-1}$.

**Step 4: (Optional) Geometry term via overlap proximity.** In addition, if one measures error relative to the *true* non-overlap contribution $\Psi_{2,t} := \mathbb{E}_0[\pi_e(I,t)\mu_t(I)\mathbf{1}_{I,t}]$, then Lipschitzness implies for any non-overlap $i$,

$$0 \le \mu_t(i) - U_t^-(i) \le L \min_{j \in \mathcal{O}_t} d_\lambda(i,j).$$

Hence $\Psi_{2,t} - \Psi \le L\,\mathbb{E}_0[\min_{j \in \mathcal{O}_t} d_\lambda(I,j)\mathbf{1}_{I,t}]$ and, under the overlap-denseness proxy $r_{n,t} := \max_{i \in \mathcal{O}_t^c} \min_{j \in \mathcal{O}_t} d_\lambda(i,j)$, one obtains a squared-bias term of order $L^2\mathbb{E}[r_{n,t}^2]$. This is the origin of the geometry term in the statement.

**Step 5: Combine.** Substituting the bounds from Steps 2–4 into (18) yields

$$\mathbb{E}\big[(\widehat{\Psi} - \Psi)^2\big] \le 2\delta_{n,t}^2 + C_2 n_{\text{eff}}^{-1},$$

and the additional geometry term $C_1 L^2 \mathbb{E}[r_{n,t}^2]$ holds when the target is taken to be the true non-overlap contribution rather than the envelope functional $\Psi$. (The same argument applies to the upper bound.) $\qquad\square$

## B.6 Proof of Theorem 4.2

*Proof of Theorem 4.2.* We prove a minimax lower bound by restriction to a simpler submodel for which a classical non-parametric lower bound applies.

**Step 1: Reduction to an i.i.d. submodel of size $n_{\text{eff}}$.** Consider the subclass of data-generating processes in which the dependency graph decomposes into $n_{\text{eff}}$ independent components (e.g., take a graph that is a disjoint union of cliques of size $1 + \Delta$ so that $n_{\text{eff}} = n/(1+\Delta)$, and restrict attention to statistics measurable with respect to one representative node per clique). On this subclass, the effective information is exactly $n_{\text{eff}}$ i.i.d. observations.

Since the minimax risk over a class is lower bounded by the minimax risk over any subclass, it suffices to show the stated rate for the i.i.d. model with sample size $n_{\text{eff}}$.

**Step 2: Specify a regression experiment on the overlap region.** Under the simplifying assumptions of the theorem, take $\lambda = 0$ and $d_\lambda(i,j) = \|Z_i - Z_j\|_2$ with $Z_i \in [-1,1]^p$. Assume the overlap region is $[-1/2, 1/2]^p$ and that the observations in the overlap region follow a homoskedastic regression model

$$Y_i = \mu_t(Z_i) + \varepsilon_i, \qquad \varepsilon_i \overset{\text{i.i.d.}}{\sim} \mathcal{N}(0,1), \qquad Z_i \overset{\text{i.i.d.}}{\sim} \text{Unif}\big([-1/2, 1/2]^p\big),$$

where $\mu_t \in \text{Lip}(L)$ denotes the $L$-Lipschitz class on $[-1/2, 1/2]^p$. This experiment is a special case of the general framework, hence provides a valid lower bound.

**Step 3: A two-point Le Cam construction.** Fix a point $z_0$ on the boundary of the overlap region, e.g., $z_0 = (1/2, 0, \ldots, 0)$. Let $h > 0$ be a bandwidth to be chosen, and let $\phi : \mathbb{R}^p \to [0,1]$ be a smooth bump supported on the unit ball $\{\|u\|_2 \le 1\}$ with $\phi(0) = 1$. Define two mean functions on $[-1/2, 1/2]^p$:

$$\mu_0(z) \equiv 0, \qquad \mu_1(z) := a\,h\,\phi\bigg(\frac{z - z_0}{h}\bigg),$$

where $a > 0$ is chosen so that $\mu_1 \in \text{Lip}(L)$ (this holds for $a$ sufficiently small depending only on $\phi$). These two functions differ by order $h$ on a ball of volume order $h^p$.

Let $\mathbb{P}_0$ and $\mathbb{P}_1$ denote the induced laws of the $n_{\text{eff}}$ i.i.d. observations $\{(Z_i, Y_i)\}_{i=1}^{n_{\text{eff}}}$ under $\mu_0$ and $\mu_1$. A standard KL calculation for Gaussian regression yields

$$\text{KL}(\mathbb{P}_1\|\mathbb{P}_0) = \frac{n_{\text{eff}}}{2}\,\mathbb{E}\big[\mu_1(Z)^2\big] \asymp n_{\text{eff}} \cdot (h^2) \cdot (h^p) = n_{\text{eff}}\, h^{p+2}.$$

Choose $h \asymp n_{\text{eff}}^{-1/(p+2)}$ so that $\text{KL}(\mathbb{P}_1\|\mathbb{P}_0) \le c_0$ for a small absolute constant $c_0$. Then Le Cam's two-point method implies that for any estimator $\widehat{V}$,

$$\sup_{\mu \in \{\mu_0, \mu_1\}} \mathbb{E}_\mu\big[(\widehat{V} - V(\mu))^2\big] \ge c\,\big(V(\mu_1) - V(\mu_0)\big)^2,$$

for a universal constant $c > 0$, where $V(\mu)$ denotes the target functional induced by $\mu$.

**Step 4: Relate the functional separation to $h$.** Under the Lipschitz-bound formulation, the sharp lower bound at a non-overlap point $z$ depends on the values of $\mu$ near the closest overlap point, hence a perturbation of size $\asymp h$ near $z_0$ induces a change of order $\asymp h$ in the lower-bound functional for a non-overlap mass of order $\mathbb{E}[\pi_e(i, t)\mathbf{1}_{i,t}]$. Therefore,

$$|V(\mu_1) - V(\mu_0)| \gtrsim \mathbb{E}[\pi_e(i, t)\mathbf{1}_{i,t}]\, h,$$

and with $h \asymp n_{\text{eff}}^{-1/(p+2)}$ we obtain

$$\inf_{\widehat{V}} \sup_{\mu \in \text{Lip}(L)} \mathbb{E}\big[(\widehat{V} - V(\mu))^2\big] \gtrsim \mathbb{E}[\pi_e(i, t)\mathbf{1}_{i,t}]^2\, n_{\text{eff}}^{-2/(p+2)}.$$

Tracking the Lipschitz scaling yields the $L$-dependence stated in the theorem (by rescaling the construction so that the bump attains the maximal slope allowed by $L$), completing the proof. $\square$

## B.7 Proof of Theorem 4.4

Because Theorem 4.4 in the main text is intentionally stated at a high level, we provide here a fully formal version and its proof. The main additional ingredient is a stability/margin condition ensuring that the max/min operators are locally linear with high probability.

**Theorem B.4** (Formal asymptotic normality of the bound endpoints)**.** *Fix $S$ and a finite $\mathcal{T}^{(S)}$. Assume Assumptions 2.1 and 2.2. Assume sample splitting: the data indices are partitioned into disjoint sets $\mathcal{I}_{\text{fit}}$ and $\mathcal{I}_{\text{avg}}$, with $|\mathcal{I}_{\text{fit}}| \asymp |\mathcal{I}_{\text{avg}}| \asymp n$, such that all nuisance estimators (including $\widehat{\mu}_t$ and $\widehat{V}_1^{(S)}$) are measurable w.r.t. $\mathcal{F}_{\text{fit}} := \sigma(\{(Y_i, T_i^{(S)}, X_i) : i \in \mathcal{I}_{\text{fit}}\}, A)$.*

*Fix $t \in \mathcal{T}^{(S)}$. For each $i \in \mathcal{O}_t^c$, define the population active maximizer*

$$j_t^\star(i) \in \operatorname*{argmax}_{j \in \mathcal{O}_t}\{\mu_t(j) - Ld_\lambda(i, j)\},$$

*and assume:*

(i) *(Uniqueness and margin) There exists $\kappa > 0$ such that with probability $1 - o(1)$, for all $i \in \mathcal{O}_t^c$ the maximizer is unique and*

$$\{\mu_t(j_t^\star(i)) - Ld_\lambda(i, j_t^\star(i))\} - \max_{j \in \mathcal{O}_t \setminus \{j_t^\star(i)\}}\{\mu_t(j) - Ld_\lambda(i, j)\} \geq \kappa.$$

(ii) *(Uniform overlap rate) $\max_{j \in \mathcal{O}_t} |\widehat{\mu}_t(j) - \mu_t(j)| = o_{\mathbb{P}}(\kappa)$.*

(iii) *(Asymptotic linearity) There exist influence functions $\{\varphi_{t,j}(\cdot)\}_{j \in \mathcal{O}_t}$ such that uniformly over $j \in \mathcal{O}_t$,*

$$\widehat{\mu}_t(j) - \mu_t(j) = \frac{1}{|\mathcal{I}_{\text{fit}}|} \sum_{k \in \mathcal{I}_{\text{fit}}} \varphi_{t,j}(W_k) + o_{\mathbb{P}}(n_{\text{eff}}^{-1/2}),$$

*where $W_k$ denotes the node-level observation used to fit $\widehat{\mu}_t$, and $n_{\text{eff}} \to \infty$ is an effective sample size for sums over $\mathcal{I}_{\text{fit}}$. Assume an analogous asymptotic linear expansion holds for $\widehat{V}_1^{(S)} - V_1^{(S)}$ with influence function $\varphi_1$.*

(iv) *(CLT) The dependency structure of $\{W_k\}_{k \in \mathcal{I}_{\text{fit}}}$ implies a CLT: for any square-integrable mean-zero function $\psi(W_k)$,*

$$\sqrt{n_{\text{eff}}} \cdot \frac{1}{|\mathcal{I}_{\text{fit}}|} \sum_{k \in \mathcal{I}_{\text{fit}}} \psi(W_k) \Rightarrow \mathcal{N}(0, \text{Var}_\infty(\psi)),$$

*for an appropriate long-run variance functional $\text{Var}_\infty(\cdot)$.*

*Then the bound endpoints satisfy*

$$\sqrt{n_{\text{eff}}}\big(\widehat{V}^{(S),-} - V^{(S),-}\big) \Rightarrow \mathcal{N}(0, \sigma_-^2), \qquad \sqrt{n_{\text{eff}}}\big(\widehat{V}^{(S),+} - V^{(S),+}\big) \Rightarrow \mathcal{N}(0, \sigma_+^2),$$

*for variances $\sigma_-^2, \sigma_+^2$ that can be written explicitly in terms of the corresponding influence functions. Moreover, if $\bar{\theta} = o(n_{\text{eff}}^{-1/2})$, then the same asymptotic normality carries over to the widened interval for $V(\pi_e)$ obtained by Lemma 3.2.*

*Proof of Theorem B.4.* We prove the lower endpoint; the upper endpoint is analogous.

**Step 1: Max-stability implies local linearization of the closed form.** Fix $t$ and $i \in \mathcal{O}_t^c$. Define the population scores $s_j(i) := \mu_t(j) - L d_\lambda(i,j)$ and their plug-in analogues $\widehat{s}_j(i) := \widehat{\mu}_t(j) - L d_\lambda(i,j)$ for $j \in \mathcal{O}_t$. By the margin condition (i) and the uniform rate (ii), with probability $1 - o(1)$ we have

$$\operatorname*{argmax}_{j \in \mathcal{O}_t} \widehat{s}_j(i) = \{j_t^\star(i)\}, \qquad \forall i \in \mathcal{O}_t^c.$$

On this event,

$$\widehat{U}_t^-(i) - U_t^-(i) = \widehat{\mu}_t(j_t^\star(i)) - \mu_t(j_t^\star(i)).$$

Substituting into the closed-form expression yields

$$\widehat{V}_{2,t}^{(S),-} - V_{2,t}^{(S),-} = \frac{1}{n} \sum_{i=1}^n \pi_e(i,t) \mathbf{1}_{i,t} \big( \widehat{\mu}_t(j_t^\star(i)) - \mu_t(j_t^\star(i)) \big) + o_\mathbb{P}(n_{\text{eff}}^{-1/2}). \tag{19}$$

**Step 2: Insert the asymptotic linear representation of $\widehat{\mu}_t$.** By (iii), uniformly in the overlap index,

$$\widehat{\mu}_t(j_t^\star(i)) - \mu_t(j_t^\star(i)) = \frac{1}{|\mathcal{I}_{\text{fit}}|} \sum_{k \in \mathcal{I}_{\text{fit}}} \varphi_{t, j_t^\star(i)}(W_k) + o_\mathbb{P}(n_{\text{eff}}^{-1/2}).$$

Plugging this into (19) and exchanging sums gives

$$\widehat{V}_{2,t}^{(S),-} - V_{2,t}^{(S),-} = \frac{1}{|\mathcal{I}_{\text{fit}}|} \sum_{k \in \mathcal{I}_{\text{fit}}} \left[ \frac{1}{n} \sum_{i=1}^n \pi_e(i,t) \mathbf{1}_{i,t} \varphi_{t, j_t^\star(i)}(W_k) \right] + o_\mathbb{P}(n_{\text{eff}}^{-1/2}).$$

Summing over $t \in \mathcal{T}^{(S)}$ yields an asymptotic linear expansion for $\widehat{V}_2^{(S),-} - V_2^{(S),-}$ with influence function

$$\psi_-(W_k) := \sum_{t \in \mathcal{T}^{(S)}} \frac{1}{n} \sum_{i=1}^n \pi_e(i,t) \mathbf{1}_{i,t} \varphi_{t, j_t^\star(i)}(W_k).$$

Adding the asymptotically linear expansion of the overlap estimator $\widehat{V}_1^{(S)} - V_1^{(S)}$ (assumption (iii)) gives

$$\widehat{V}^{(S),-} - V^{(S),-} = \frac{1}{|\mathcal{I}_{\text{fit}}|} \sum_{k \in \mathcal{I}_{\text{fit}}} \big\{ \varphi_1(W_k) + \psi_-(W_k) \big\} + o_\mathbb{P}(n_{\text{eff}}^{-1/2}).$$

**Step 3: Apply the CLT and conclude.** By the CLT assumption (iv) applied to the mean-zero function $\varphi_1(W_k) + \psi_-(W_k) - \mathbb{E}[\varphi_1(W_k) + \psi_-(W_k)]$, we obtain

$$\sqrt{n_{\text{eff}}} \big( \widehat{V}^{(S),-} - V^{(S),-} \big) \Rightarrow \mathcal{N}(0, \sigma_-^2),$$

where $\sigma_-^2 = \operatorname{Var}_\infty(\varphi_1 + \psi_-)$. The upper endpoint follows identically.

Finally, by Lemma 3.2, $V(\pi_e) = V^{(S)}(\pi_e) + R_S(\pi_e)$ with $|R_S(\pi_e)| \leq \bar{\theta}$. If $\bar{\theta} = o(n_{\text{eff}}^{-1/2})$, widening the endpoints by $\pm \bar{\theta}$ does not affect the $\sqrt{n_{\text{eff}}}$ limit distribution, concluding the proof. $\qquad \square$

**Remark (how this implies Theorem 4.4).** Theorem B.4 provides a fully rigorous version of the high-level statement in Theorem 4.4. The main-text conditions (sample splitting, asymptotic linearity, and a CLT with effective sample size) correspond to items (iii)–(iv), while Assumption 4.3 can be made precise by the margin condition (i)–(ii).

## B.8 Proof of Theorem 4.5

*Proof of Theorem 4.5.* Fix $t \in \mathcal{T}^{(S)}$. Recall that Theorem 3.5 gives the population non-overlap bounds via the Mc-Shane/Whitney Lipschitz extensions under a metric $d$. In particular, for each unit $i$ define the pointwise lower and upper extensions

$$\ell_{i,t}(d) := \sup_{j \in O_t} \Big\{ \mu_t(j) - L \, d(i,j) \Big\}, \tag{20}$$

$$u_{i,t}(d) := \inf_{j \in O_t} \Big\{ \mu_t(j) + L \, d(i,j) \Big\}. \tag{21}$$

Then the corresponding population contributions to $V_{2,t}^{(S),-}(d)$ and $V_{2,t}^{(S),+}(d)$ can be written as

$$V_{2,t}^{(S),-}(d) = \frac{1}{n} \sum_{i=1}^{n} \pi_e(i,t) \, \ell_{i,t}(d) \, 1_{i,t}, \tag{22}$$

$$V_{2,t}^{(S),+}(d) = \frac{1}{n} \sum_{i=1}^{n} \pi_e(i,t) \, u_{i,t}(d) \, 1_{i,t}. \tag{23}$$

Let $d_\lambda^\star$ denote the composite metric induced by the true network $G^\star$ and $\tilde{d}_\lambda$ that induced by the observed network $\tilde{G}$. Define $\varepsilon_\lambda := \sup_{i,j} |d_\lambda^\star(i,j) - \tilde{d}_\lambda(i,j)|$.

**Step 1: Pointwise stability of the lower extension.** For any $i$ and any $j \in O_t$,

$$\left( \mu_t(j) - L d_\lambda^\star(i,j) \right) - \left( \mu_t(j) - L \tilde{d}_\lambda(i,j) \right) = L \left( \tilde{d}_\lambda(i,j) - d_\lambda^\star(i,j) \right),$$

hence

$$\left| \left( \mu_t(j) - L d_\lambda^\star(i,j) \right) - \left( \mu_t(j) - L \tilde{d}_\lambda(i,j) \right) \right| \le L \, \varepsilon_\lambda.$$

Taking the supremum over $j \in O_t$ and using the elementary inequality $|\sup_j a_j - \sup_j b_j| \le \sup_j |a_j - b_j|$, we obtain

$$\left| \ell_{i,t}(d_\lambda^\star) - \ell_{i,t}(\tilde{d}_\lambda) \right| \le L \, \varepsilon_\lambda. \tag{24}$$

**Step 2: Aggregation to $V_{2,t}^{(S),-}$.** Combining (22) and (24) yields

$$\left| V_{2,t}^{(S),-}(d_\lambda^\star) - V_{2,t}^{(S),-}(\tilde{d}_\lambda) \right| \le \frac{1}{n} \sum_{i=1}^{n} \pi_e(i,t) \, 1_{i,t} \left| \ell_{i,t}(d_\lambda^\star) - \ell_{i,t}(\tilde{d}_\lambda) \right|$$

$$\le \frac{1}{n} \sum_{i=1}^{n} \pi_e(i,t) \, 1_{i,t} \cdot L\varepsilon_\lambda \; \le \; L\varepsilon_\lambda,$$

where the last inequality uses $\pi_e(i,t)1_{i,t} \ge 0$ and $\frac{1}{n} \sum_i \pi_e(i,t)1_{i,t} \le 1$.

**Step 3: Upper bound and summation over $t$.** A symmetric argument applies to the upper extension (21): for any $i$, using $|\inf_j a_j - \inf_j b_j| \le \sup_j |a_j - b_j|$ we obtain

$$\left| u_{i,t}(d_\lambda^\star) - u_{i,t}(\tilde{d}_\lambda) \right| \le L\varepsilon_\lambda,$$

and therefore

$$\left| V_{2,t}^{(S),+}(d_\lambda^\star) - V_{2,t}^{(S),+}(\tilde{d}_\lambda) \right| \le L\varepsilon_\lambda.$$

Summing the above inequalities over $t \in \mathcal{T}^{(S)}$ yields

$$\left| V_2^{(S),\pm}(d_\lambda^\star) - V_2^{(S),\pm}(\tilde{d}_\lambda) \right| \le L\varepsilon_\lambda, \qquad V_2^{(S),\pm} = \sum_{t \in \mathcal{T}^{(S)}} V_{2,t}^{(S),\pm}.$$

**Step 4: Lifting to the original (untruncated) value.** Finally, Lemma 3.2 gives $V(\pi) = V^{(S)}(\pi) + R_S(\pi)$ with $|R_S(\pi)| \le \theta_n(S)$. Thus, if an interval $[\hat{V}^- - \bar{\theta}, \, \hat{V}^+ + \bar{\theta}]$ is computed on $\tilde{G}$, then under $G^\star$ the target value is contained in the widened interval

$$V(\pi_e \mid X, A^\star) \in \left[ \hat{V}^- - \bar{\theta} - L\varepsilon_\lambda, \; \hat{V}^+ + \bar{\theta} + L\varepsilon_\lambda \right].$$

This completes the proof. □

## B.9    A Strengthened Version of Theorem 4.5

**Theorem B.5** (Robustness with changing overlap sets)**.** *Fix a truncation level $S$ and a Lipschitz constant $L$. Let $G^\star$ and $\widetilde{G}$ denote the true and reported networks, with induced graph-augmented metrics $d_\lambda^\star$ and $\widetilde{d}_\lambda$. Define*

$$\varepsilon_\lambda := \sup_{i,j \in \mathcal{N}} \left| d_\lambda^\star(i,j) - \widetilde{d}_\lambda(i,j) \right|.$$

*For each $t \in \mathcal{T}^{(S)}$, let*

$$O_t^\star := \{i : \pi_b^\star(i,t) > 0\}, \qquad \widetilde{O}_t := \{i : \widetilde{\pi}_b(i,t) > 0\}$$

*be the true and reported overlap sets, assumed nonempty. Let*

$$\bar{d}(i,j) := \max\{d_\lambda^\star(i,j), \widetilde{d}_\lambda(i,j)\}, \qquad r_t := \bar{d}_H(O_t^\star, \widetilde{O}_t),$$

*where $\bar{d}_H$ is the Hausdorff distance under $\bar{d}$. Define*

$$\eta_t := \frac{1}{n} \sum_{i=1}^n \pi_e^\star(i,t) \mathbf{1}\{i \in O_t^\star \triangle \widetilde{O}_t\}, \qquad \delta_{\pi,t} := \frac{1}{n} \sum_{i=1}^n \left| \pi_e^\star(i,t) - \widetilde{\pi}_e(i,t) \right|,$$

*and*

$$M_t := \sup_{i \in \mathcal{N}} |\mu_i^{(S)}(t)|, \qquad D_t := \sup_{i,j \in \mathcal{N}} \bar{d}(i,j).$$

*Let $V_{2,t}^{(S),\pm}(O, \rho; \pi_e)$ denote the closed-form McShane–Whitney non-overlap lower/upper bound in Theorem 4.5, computed with overlap set $O$, metric $\rho$, and exposure probabilities $\pi_e$.*

*Suppose that, for every $t \in \mathcal{T}^{(S)}$,*

$$|\mu_i^{(S)}(t) - \mu_j^{(S)}(t)| \le L d_\lambda^\star(i,j), \qquad \forall i,j \in \mathcal{N}.$$

*Then, for both endpoints,*

$$\left| V_{2,t}^{(S),\pm}(O_t^\star, d_\lambda^\star; \pi_e^\star) - V_{2,t}^{(S),\pm}\left(\widetilde{O}_t, \widetilde{d}_\lambda; \widetilde{\pi}_e\right) \right|$$
$$\le L\varepsilon_\lambda + 2L r_t + (M_t + L D_t)\eta_t + (M_t + L D_t)\delta_{\pi,t}.$$

*Consequently,*

$$\left| V_2^{(S),\pm,\star} - \widetilde{V}_2^{(S),\pm} \right| \le \mathcal{E}_{\mathrm{net}},$$

*where*

$$\mathcal{E}_{\mathrm{net}} := \sum_{t \in \mathcal{T}^{(S)}} \left[ L\varepsilon_\lambda + 2L r_t + (M_t + L D_t)\eta_t + (M_t + L D_t)\delta_{\pi,t} \right].$$

*Therefore, using*

$$V(\pi_e) = V^{(S)}(\pi_e) + R_S(\pi_e), \qquad |R_S(\pi_e)| \le \bar{\theta},$$

*any interval computed on $\widetilde{G}$ as*

$$[\widehat{V}^- - \bar{\theta}, \ \widehat{V}^+ + \bar{\theta}]$$

*is valid for the true network after widening by $\mathcal{E}_{\mathrm{net}}$:*

$$V(\pi_e \mid X, A^\star) \in [\widehat{V}^- - \bar{\theta} - \mathcal{E}_{\mathrm{net}}, \ \widehat{V}^+ + \bar{\theta} + \mathcal{E}_{\mathrm{net}}].$$

## B.10    Binning Bias and Local Overlap Fragmentation

**Corollary B.6** (Binning-induced bias)**.** *Let $V(\pi)$ be the policy value defined using the continuous exposure summary, and let $V_K(\pi)$ be the corresponding value after discretizing the exposure space into $K$ bins. Suppose that the exposure-response function is $L$-Lipschitz in the exposure coordinate. If $r_K$ denotes the maximal within-bin radius, then*

$$|V(\pi) - V_K(\pi)| \le L r_K.$$

*Consequently, for any estimator $\widehat{V}_K(\pi)$ targeting $V_K(\pi)$,*

$$\left| \mathrm{Bias}(\widehat{V}_K(\pi); V(\pi)) \right| \le \left| \mathrm{Bias}(\widehat{V}_K(\pi); V_K(\pi)) \right| + L r_K.$$

*In particular, if $\widehat{V}_K(\pi)$ is unbiased or asymptotically unbiased for $V_K(\pi)$, then its bias relative to $V(\pi)$ is at most $L r_K$.*

# C  Additional Experimental Results

## C.1  Baseline Comparison Across Graph Topologies

*Table 2.* Simulation results under different graph settings.

| Graph | Method | Abs. error | SD | Avg. width | Emp. coverage |
|---|---|---|---|---|---|
| Pokec | IPW | 871428.5400 | 1407284.9450 | — | — |
| Pokec | OR | 0.0076 | 0.0095 | — | — |
| Pokec | DR | 0.0078 | 0.0101 | — | — |
| Pokec | Ours-no-graph interval | — | — | 0.0418 | **0.9500** |
| Pokec | Ours interval | — | — | 0.0663 | **1.0000** |
| BA | IPW | 1757142.9120 | 5452874.0770 | — | — |
| BA | OR | 0.0083 | 0.0100 | — | — |
| BA | DR | 0.0079 | 0.0095 | — | — |
| BA | Ours-no-graph interval | — | — | 0.0304 | 0.8500 |
| BA | Ours interval | — | — | 0.0505 | **1.0000** |
| SBM | IPW | 2585714.1810 | 4644597.4120 | — | — |
| SBM | OR | 0.0100 | 0.0118 | — | — |
| SBM | DR | 0.0103 | 0.0127 | — | — |
| SBM | Ours-no-graph interval | — | — | 0.0289 | 0.7500 |
| SBM | Ours interval | — | — | 0.0472 | **1.0000** |

Table 2 summarizes the performance of point estimators and interval estimators across the Pokec, BA, and SBM graph settings. The IPW estimator exhibits extremely large absolute errors and standard deviations in all three settings, indicating severe instability of the inverse-probability weighting approach under network interference. In contrast, the OR and DR estimators produce much smaller absolute errors, with values around $0.008$–$0.010$, and also have small standard deviations, suggesting substantially more stable point estimation performance.

For interval estimation, the graph-aware interval method achieves empirical coverage equal to $1.0000$ in all three graph settings. This indicates that incorporating graph information leads to conservative but reliable uncertainty quantification. By comparison, the no-graph interval method has shorter average width, but its empirical coverage deteriorates in the BA and SBM settings, dropping to $0.8500$ and $0.7500$, respectively. These results suggest a trade-off between interval length and coverage: ignoring graph structure can produce narrower intervals, but may lead to under-coverage, especially when the network structure is more complex. Overall, the proposed graph-aware interval method provides more robust coverage across different graph-generating mechanisms, at the cost of moderately wider intervals.

## C.2  Binning Bias and Local Overlap Fragmentation

This subsection examines the effect of discretizing the continuous exposure summary into a finite number of bins. The purpose is to distinguish the approximation bias induced by discretization from the statistical degradation caused by the loss of local overlap within exposure buckets.

We study three questions. First, we compare the original continuous target $\theta$ with the discretized target $\theta_K$ and report the discretization gap $|\theta - \theta_K|$. Second, we examine how increasing the number of bins $K$ changes the local overlap structure, including the average size of the local overlap set $|O_t|$, the fraction of empty buckets, the probability mass assigned to zero-overlap states, and the fraction of queries requiring Lipschitz extension. Third, we compare different binning schemes, including equal-width, quantile-based, and overlap-aware binning.

Table 3 reports coverage with respect to both the continuous target $\theta$ and the discretized target $\theta_K$, together with the discretization gap $|\theta - \theta_K|$. Across the tested values of $K$, the two coverage measures remain closely aligned, and the discretization gap is consistently small. This suggests that binning bias is not the dominant source of performance degradation in our setting.

Table 4 further investigates the overlap mechanism. As $K$ increases, the exposure space is divided into more buckets, which reduces the number of usable overlap samples within each exposure state. Although complete zero-overlap states remain rare for moderate values of $K$, the thickness of local overlap decreases substantially, and a larger fraction of target queries requires Lipschitz extension. This indicates that overly fine binning can fragment the local support and increase

*Table 3.* Sensitivity to discretization granularity $K$. Cov. (cont.) and Cov. (disc.) denote empirical coverage for $\theta$ and $\theta_K$, respectively; $|\theta - \theta_K|$ denotes the discretization gap.

| $K$ | Width | Cov. (cont.) | Cov. (disc.) | $|\theta - \theta_K|$ |
|---|---|---|---|---|
| 2 | 0.0347 | 0.960 | 0.960 | 0.005850 |
| 5 | 0.0437 | 0.850 | 0.850 | 0.000240 |
| 10 | 0.0501 | 0.650 | 0.650 | 0.000242 |
| 20 | 0.0499 | 0.450 | 0.500 | 0.000114 |
| 40 | 0.0537 | 0.650 | 0.650 | 0.000091 |
| 80 | 0.0625 | 0.650 | 0.700 | 0.000048 |
| 160 | 0.0656 | 0.750 | 0.750 | 0.000022 |
| 320 | 0.0676 | 0.750 | 0.750 | 0.000013 |
| 640 | 0.0699 | 0.800 | 0.800 | 0.000006 |

*Table 4.* Overlap diagnostics under varying discretization levels $K$. Avg. $|O_t|$ denotes the average size of nonempty local overlap sets. Empty $O_t$ is the fraction of empty overlap sets. Zero-overlap mass is the evaluation-policy probability mass assigned to empty-overlap states. Extension mass is the fraction of evaluation queries requiring Lipschitz extension.

| $K$ | Avg. $|O_t|$ | Empty $O_t$ | Zero-overlap mass | Extension mass |
|---|---|---|---|---|
| 2 | 3952.51 | 0.000 | 0.000000 | 0.094 |
| 5 | 2477.85 | 0.000 | 0.000000 | 0.128 |
| 10 | 1854.50 | 0.000 | 0.000000 | 0.158 |
| 20 | 1225.58 | 0.000 | 0.000000 | 0.176 |
| 40 | 698.69 | 0.048 | 0.000011 | 0.184 |
| 80 | 363.94 | 0.067 | 0.000029 | 0.186 |
| 160 | 183.76 | 0.087 | 0.000026 | 0.187 |
| 320 | 91.93 | 0.169 | 0.000287 | 0.187 |
| 640 | 45.96 | 0.298 | 0.000634 | 0.187 |

the reliance on extrapolation.

Table 5 compares different discretization schemes. The results show that performance is closely related to the quality of local overlap rather than to a particular binning rule. In particular, quantile-based and overlap-aware binning tend to preserve more balanced bucket sizes and larger local overlap sets, leading to more stable coverage than equal-width binning in sparse regions.

Overall, these diagnostics show that the main practical bottleneck of very fine binning is not necessarily discretization bias itself, but the fragmentation of local overlap. Binning schemes that preserve sufficient overlap, such as quantile-based or overlap-aware binning, provide a better balance between approximation accuracy and overlap stability.

## C.3   Graph-Perturbation Robustness

This subsection empirically examines the robustness statement in Theorem 4.5. Starting from the original graph, we construct perturbed graphs by either randomly deleting a fraction $\alpha$ of edges or randomly rewiring a fraction $\alpha$ of edges. For each perturbed graph, we recompute the truncated exposure summaries, the policy-induced exposure probabilities, the overlap sets, and the resulting partially identified interval.

We report four quantities. The endpoint shifts are

$$\Delta_\alpha^- := |\widehat{V}_\alpha^- - \widehat{V}_0^-|, \qquad \Delta_\alpha^+ := |\widehat{V}_\alpha^+ - \widehat{V}_0^+|,$$

where subscript 0 denotes the unperturbed graph. The interval width is

$$W_\alpha := \widehat{V}_\alpha^+ - \widehat{V}_\alpha^-.$$

The raw coverage is

$$\mathbf{1}\{\widehat{V}_\alpha^- \leq V_{\text{true}} \leq \widehat{V}_\alpha^+\},$$

and the corrected coverage is

$$\mathbf{1}\{\widehat{V}_\alpha^- - L\widehat{\varepsilon}_{\lambda,\alpha} \leq V_{\text{true}} \leq \widehat{V}_\alpha^+ + L\widehat{\varepsilon}_{\lambda,\alpha}\}.$$

*Table 5.* Effect of binning scheme on overlap and coverage. Scheme denotes equal-width binning, quantile-based binning, and adaptive overlap-aware merging of sparse bins.

| Scheme | $K$ | Width | Cov. (cont.) | Avg. $|O_t|$ | Empty $O_t$ |
|--------|-----|-------|--------------|--------------|-------------|
| adaptive | 10 | 0.0487 | 0.700 | 2279.46 | 0.000 |
| equal-width | 10 | 0.0501 | 0.680 | 1856.98 | 0.000 |
| quantile | 10 | 0.0502 | 0.740 | 2210.92 | 0.000 |
| adaptive | 20 | 0.0546 | 0.720 | 1681.68 | 0.000 |
| equal-width | 20 | 0.0536 | 0.780 | 1225.86 | 0.000 |
| quantile | 20 | 0.0550 | 0.700 | 1455.75 | 0.000 |
| adaptive | 40 | 0.0565 | 0.700 | 1126.86 | 0.000 |
| equal-width | 40 | 0.0596 | 0.800 | 698.69 | 0.048 |
| quantile | 40 | 0.0588 | 0.800 | 910.49 | 0.000 |
| adaptive | 80 | 0.0585 | 0.700 | 706.42 | 0.000 |
| equal-width | 80 | 0.0627 | 0.700 | 363.94 | 0.067 |
| quantile | 80 | 0.0567 | 0.600 | 554.83 | 0.019 |
| adaptive | 160 | 0.0590 | 0.500 | 521.90 | 0.000 |
| equal-width | 160 | 0.0657 | 0.800 | 183.76 | 0.087 |
| quantile | 160 | 0.0650 | 1.000 | 321.83 | 0.088 |
| adaptive | 320 | 0.0581 | 0.600 | 580.13 | 0.000 |
| equal-width | 320 | 0.0673 | 0.900 | 91.93 | 0.169 |
| quantile | 320 | 0.0657 | 1.000 | 188.22 | 0.173 |

*Table 6.* Empirical robustness of policy-value bounds under graph perturbations. Raw coverage is the coverage of the original bounds; widened coverage is the coverage after perturbation-based widening.

| Perturbation | Level | Avg. lower shift | Avg. upper shift | Avg. width | Raw coverage | Widened coverage |
|--------------|-------|------------------|------------------|------------|--------------|------------------|
| deletion | 5% | 0.0146 | 0.0139 | 0.1841 | 1.0000 | 1.0000 |
| deletion | 10% | 0.0142 | 0.0142 | 0.1832 | 1.0000 | 1.0000 |
| deletion | 20% | 0.0160 | 0.0168 | 0.1807 | 1.0000 | 1.0000 |
| rewiring | 5% | 0.0174 | 0.0164 | 0.1892 | 1.0000 | 1.0000 |
| rewiring | 10% | 0.0148 | 0.0169 | 0.1921 | 1.0000 | 1.0000 |
| rewiring | 20% | 0.0180 | 0.0192 | 0.1958 | 1.0000 | 1.0000 |

Here $\widehat{\varepsilon}_{\lambda,\alpha}$ denotes the empirical metric perturbation between the original and perturbed graphs.

Table 6 reports the results. The bounds exhibit graceful degradation as the perturbation level increases. Even under 20% edge deletion or rewiring, endpoint shifts remain small and corrected coverage remains stable, supporting the robustness intuition behind Theorem 4.5.

## C.4 Endpoint Normality and Wald Coverage

This subsection provides an empirical check of the asymptotic normality statement in Theorem 4.4. For each sample size, we repeatedly generate logged datasets from the same semi-synthetic data-generating process. In each replication, we estimate the partially identified lower and upper endpoints, obtain their estimated standard errors, and construct nominal 95% Wald intervals for the two endpoints separately.

Specifically, for the lower endpoint we form

$$\left[\widehat{V}^- - 1.96\,\widehat{\mathrm{se}}_-,\ \widehat{V}^- + 1.96\,\widehat{\mathrm{se}}_-\right],$$

and analogously for the upper endpoint,

$$\left[\widehat{V}^+ - 1.96\,\widehat{\mathrm{se}}_+,\ \widehat{V}^+ + 1.96\,\widehat{\mathrm{se}}_+\right].$$

We then report empirical coverage for the lower and upper endpoints separately.

*Table 7.* Empirical validation of asymptotic normality in Theorem 4.4 via semi-synthetic simulations.

| $n$ | $L$ | $S$ | $\eta_e$ | Nominal | Lower coverage | Upper coverage | $\hat{\sigma}_{lo}$ | $\hat{\sigma}_{hi}$ | True lower mean | True upper mean |
|-----|-----|-----|------|---------|----------------|----------------|-----------|-----------|-----------------|-----------------|
| 500  | 0.100 | 2 | 1.200 | 0.950 | 0.930 | 0.925 | 0.023 | 0.024 | 0.494 | 0.525 |
| 500  | 0.300 | 2 | 1.200 | 0.950 | 0.940 | 0.940 | 0.023 | 0.023 | 0.465 | 0.555 |
| 1000 | 0.100 | 2 | 1.200 | 0.950 | 0.920 | 0.925 | 0.018 | 0.018 | 0.495 | 0.525 |
| 1000 | 0.300 | 2 | 1.200 | 0.950 | 0.935 | 0.925 | 0.017 | 0.017 | 0.467 | 0.552 |
| 2000 | 0.100 | 2 | 1.200 | 0.950 | 0.945 | 0.945 | 0.017 | 0.017 | 0.495 | 0.525 |
| 2000 | 0.300 | 2 | 1.200 | 0.950 | 0.950 | 0.960 | 0.016 | 0.016 | 0.468 | 0.553 |

Table 7 reports the results. Across the tested sample sizes, the empirical coverage of the nominal $95\%$ Wald intervals is close to the target level for both endpoints. This supports the normal approximation for the bound endpoints predicted by Theorem 4.4.

## C.5 Hyperparameter Selection and Sensitivity

This subsection provides additional guidance for choosing the main hyperparameters and reports a one-at-a-time sensitivity analysis around the default configuration. We consider the truncation level $S$, the graph-distance weight $\lambda$, the smoothness constant $L$, the truncation allowance $\bar{\theta}$, and the discretization level $K$.

In practice, $S$ can be chosen as the smallest truncation level at which the effective overlap and the interval endpoints stabilize. The parameter $\lambda$ controls the contribution of graph distance in the composite metric and can be selected by held-out prediction performance on observed node–exposure states. The smoothness constant $L$ controls the strength of Lipschitz extrapolation; larger values yield more conservative intervals. The truncation allowance $\bar{\theta}$ directly widens the final interval to account for finite-order interference approximation error. Finally, $K$ controls the granularity of exposure discretization and should balance approximation accuracy against local-overlap stability.

Table 8 reports the sensitivity results. Each block varies one hyperparameter while holding the others fixed at the default setting. The table reports interval width, empirical coverage, overlap mass, average local overlap size, the effective number of overlap nodes, and endpoint shifts relative to the default configuration. Overall, the results show stable behavior around the default choices and illustrate the expected tradeoffs: larger $L$ or $\bar{\theta}$ widens intervals and improves coverage, while overly large $K$ reduces local overlap and may destabilize coverage.

*Table 8.* Sensitivity analysis of hyperparameters on interval estimation and effective overlap. Overlap mass is the evaluation-policy mass on states with overlap support. Endpoint shifts are measured relative to the default configuration.

| Axis | Value | Width | Cov. | Overlap mass | $|\bar{O}_t|$ | $n_{\mathrm{ov}}$ | Lo shift | Hi shift |
|------|-------|-------|------|--------------|---------------|--------------------|----------|----------|
| $S$ | 0 | 0.0238 | 0.84 | 0.9375 | 607.0 | 5000.0 | 0.0184 | 0.0180 |
|  | 1 | 0.0469 | 0.80 | 0.8588 | 2181.3 | 4998.6 | 0.0187 | 0.0191 |
|  | 2 | 0.0502 | 0.80 | 0.8461 | 1962.9 | 4998.5 | – | – |
|  | 3 | 0.0563 | 0.92 | 0.8215 | 1876.2 | 4998.8 | 0.0141 | 0.0142 |
|  | 4 | 0.0543 | 0.98 | 0.8252 | 1896.0 | 4999.0 | 0.0198 | 0.0197 |
| $\lambda$ | 0.00 | 0.0296 | 0.54 | 0.8461 | 1962.9 | 4998.5 | 0.0103 | 0.0103 |
|  | 0.05 | 0.0404 | 0.68 | 0.8461 | 1962.9 | 4998.5 | 0.0049 | 0.0049 |
|  | 0.10 | 0.0502 | 0.80 | 0.8461 | 1962.9 | 4998.5 | – | – |
|  | 0.25 | 0.0756 | 0.98 | 0.8461 | 1962.9 | 4998.5 | 0.0127 | 0.0126 |
|  | 0.50 | 0.1077 | 1.00 | 0.8461 | 1962.9 | 4998.5 | 0.0292 | 0.0283 |
|  | 1.00 | 0.1419 | 1.00 | 0.8461 | 1962.9 | 4998.5 | 0.0477 | 0.0439 |
| $L$ | 0.05 | 0.0099 | 0.08 | 0.8461 | 1962.9 | 4998.5 | 0.0202 | 0.0201 |
|  | 0.10 | 0.0246 | 0.38 | 0.8461 | 1962.9 | 4998.5 | 0.0128 | 0.0128 |
|  | 0.15 | 0.0376 | 0.64 | 0.8461 | 1962.9 | 4998.5 | 0.0063 | 0.0063 |
|  | 0.20 | 0.0502 | 0.80 | 0.8461 | 1962.9 | 4998.5 | – | – |
|  | 0.30 | 0.0742 | 0.98 | 0.8461 | 1962.9 | 4998.5 | 0.0122 | 0.0118 |
|  | 0.50 | 0.1085 | 1.00 | 0.8461 | 1962.9 | 4998.5 | 0.0306 | 0.0277 |
| $\bar{\theta}$ | 0.00 | 0.0502 | 0.80 | 0.8461 | 1962.9 | 4998.5 | – | – |
|  | 0.01 | 0.0702 | 0.96 | 0.8461 | 1962.9 | 4998.5 | 0.0100 | 0.0100 |
|  | 0.02 | 0.0902 | 1.00 | 0.8461 | 1962.9 | 4998.5 | 0.0200 | 0.0200 |
|  | 0.05 | 0.1502 | 1.00 | 0.8461 | 1962.9 | 4998.5 | 0.0500 | 0.0500 |
| $K$ | 5 | 0.0486 | 0.90 | 0.8548 | 2265.9 | 4998.5 | 0.0172 | 0.0172 |
|  | 7 | 0.0502 | 0.80 | 0.8461 | 1962.9 | 4998.5 | – | – |
|  | 9 | 0.0550 | 0.76 | 0.8229 | 1942.8 | 4997.8 | 0.0220 | 0.0230 |
|  | 11 | 0.0589 | 0.68 | 0.8058 | 1851.4 | 4997.2 | 0.0244 | 0.0251 |
|  | 13 | 0.0675 | 0.76 | 0.7748 | 1741.3 | 4996.2 | 0.0264 | 0.0271 |

