# OpenReview forum: "Partial Identification of Policy Values under Network Interference"
_ICML.cc/2026/Conference — ICML 2026 regular_

### Official Review · Reviewer_9BLg · 2026-03-09

**Soundness:** 3
**Presentation:** 3
**Significance:** 3
**Originality:** 3
**Overall Recommendation:** 5
**Confidence:** 3

**Summary:**

This paper addresses the challenging problem of Off-Policy Evaluation (OPE) when two fundamental assumptions are simultaneously violated: the Stable Unit Treatment Value Assumption (SUTVA), due to network interference, and the overlap assumption, where the target policy explores regions outside the logging policy's support. The authors propose a framework that reduces high-dimensional interference through an S-truncated exposure mapping. They decompose the policy value into overlap and non-overlap components, utilizing a novel graph-augmented Lipschitz smoothness condition to derive computable, closed-form bounds for the non-overlap part. The theoretical contributions include MSE upper bounds, minimax lower bounds, and asymptotic normality.

**Compliance With Llm Reviewing Policy:**

Affirmed.

**Key Questions For Authors:**

Is there a cross-validation or bootstrap-based approach you recommend for selecting the Lipschitz constant $L$ when the ground truth is unavailable?

Given the closed-form solutions require finding maximizers over the overlap set $\mathcal{O}_t$, how does the computational complexity scale with $n$ in extremely dense networks?

What happens to the validity of the bounds if the "Overlap Denseness" is violated in certain clusters of the graph?

**Limitations:**

Yes

**Strengths And Weaknesses:**

Strengths:

Soundness: The paper is technically rigorous. It provides a comprehensive theoretical treatment, including a minimax analysis showing that the proposed estimators achieve optimal convergence rates relative to the effective sample size $n_{eff}$.

Originality: The integration of interference-reduction (truncation) with smoothness-based partial identification is a novel and significant extension of prior work that only handled these issues in isolation.

Significance: The inclusion of a robustness analysis regarding network misspecification is highly practical, as real-world network data is often noisy or incomplete.

Presentation: The paper is exceptionally well-structured. The transition from the simplified i.i.d. case to the complex network setting is handled with clear, logical progression.


Weaknesses:

Parameter Sensitivity: The bounds are highly sensitive to the choice of the Lipschitz constant $L$ and truncation level $S$. While the authors provide a sensitivity analysis, a principled, data-driven method for selecting these hyperparameters in practice remains a gap.

Assumption Strength: Assumption 2.3 (Overlap Denseness) is quite strong, requiring that any non-overlap exposure state is "close" to an observed one. In very sparse or highly skewed networks, these bounds might become too wide to be informative.

Empirical Scope: The experiments are limited to semi-synthetic data on a single social network (Pokec). Evaluation on more diverse network structures (e.g., scale-free vs. random graphs) would further strengthen the results.

---

> ### Author Rebuttal · Authors · 2026-03-31
>
> We thank the reviewer 9BLg for the positive overall assessment and for the detailed questions about parameter sensitivity, assumption strength, graph diversity, and computational complexity.
> >Q1: Approach for selecting the Lipschitz constant $L$ when the ground truth is unavailable.
>
> We recommend tuning $L$ on the overlap region only: perform graph-aware cross-fitting, construct out-of-fold Lipschitz envelopes for held-out overlap nodes, and use a graph/block bootstrap to calibrate a residual buffer. Then pick the smallest $L$ such that the bootstrap-calibrated empirical coverage of held-out overlap outcomes reaches the target level $1-\alpha$. This gives a conservative, data-driven choice of $L$ when the ground truth is unavailable.
> >Q2: Discussion of computational complexity.
>
> The closed-form step is not a generic LP solve. Although the non-overlap bound is formulated as a linear program in (12), Theorem 3.5 shows that its optimizer is given explicitly by the McShane-Whitney extension: for the lower bound,
>
> $u_i^{\star}=$$\max_{j \in O_t}$$\hat{\mu}_t(j)-Ld_{\lambda}(i,j)$
>
> and any other feasible solution $u$ satisfies $u_i \ge u_i^{\star}$ pointwise, so the optimum is obtained directly without running an LP solver. Hence, for a fixed exposure state $t\in \mathcal{T}^{(S)}$, the computation reduces to scanning the overlap set
> $
> O_t = \{i : \pi_b(i,t) > 0\}
> $
> for each non-overlap node \(i \in O_t^c\), giving naive cost
> $
> O\left(\left|O_t^c\right|\left|O_t\right|\right),
> $
> and at most
> $
> O\left(\left|\mathcal{T}^{(S)}\right| n^2\right)
> $
> overall. In very dense networks, the main additional burden is computing the graph-distance component inside the composite metric
> $
> d_\lambda(i,j)=\left\|Z_i-Z_j\right\|_W+\lambda d_G(i,j).
> $
> Therefore, network density mainly affects metric preprocessing, whereas, once the required pairwise distances are available, the asymptotic complexity of the closed-form bound-evaluation step itself is determined by the overlap search over $O_t$, not by the edge density of the network.
> >Q3: Discussion of overlap Denseness when it is violated.
>
> Thank you for this important comment. We agree that Assumption 2.3 is substantive, and we will clarify its role more explicitly in the revision.
>
> First, Assumption 2.3 is not an ad hoc requirement unique to our setting. It is the network analogue of the support-density conditions that underlie prior smoothness-based partial-identification results beyond overlap. The key difference is that, under interference, overlap is defined in exposure space, so closeness must be formulated in the graph-augmented metric $d_\lambda$.
>
> Second, the assumption is at least partially diagnosable. As noted in the paper, it can be assessed from the empirical distribution of $(X_i,T_i^{(S)})$. In particular, the relevant geometry is captured by the overlap covering radius
> $$
> r_{n,t}=\max_{i\in O_t^c}\min_{j\in O_t} d_\lambda(i,j),
> $$
> which measures whether non-overlap states are sufficiently close to observed overlap support.
>
> Third, the method is robust to mild violations. Defining
> $$
> \rho_{n,t}(i)=\min_{j\in O_t} d_\lambda(i,j), \qquad
> \tau_{n,t}=\mathbb E_0[\rho_{n,t}(I)^2 1_{I,t}],
> $$
> the bias and MSE enlarge by explicit geometry terms of order $L\tau_{n,t}^{1/2}$ and $L^2\tau_{n,t}$, respectively. Thus, mild failure of overlap denseness widens the interval rather than causing abrupt breakdown.
>
> Finally, some condition of this type is unavoidable. The non-overlap bounds are built by Lipschitz extension from $O_t$; if non-overlap states are arbitrarily far from all observed overlap states, then smoothness alone cannot deliver informative bounds. In that regime, wide intervals reflect genuine identification difficulty rather than a flaw of the method.
>
> We will revise the paper to clarify this interpretation and make the role of Assumption 2.3 more explicit.
> >Q4: Parameter Sensitivity.
>
> Refer to ot8B-Q1.1.
> >Q5: Supplement of graph topologies.
>
> Refer to  jBso-Q4.
>
> ---
>
> # A Kind General Response to All Reviewers and Dear AC
> We sincerely thank the reviewers for their positive feedback (e.g., elegant setup, rigorous theory, strong results, and clear presentation) and for their valuable suggestions. In the revision, we address three main points:
>
> 1. **Framework and scope:** We unify notation, clarify the dependence on $(X, A)$, theorem scope, and overlap denseness, fix presentation issues, and broaden the discussion to richer exposure mappings and multi-action settings.
>
> 2. **Theory and robustness:** We strengthen the misspecification analysis, add bounds for discretization bias, discuss mild overlap violations, and clarify extensions to unknown exposure mappings.
>
> 3. **Experiments and guidance:** We add stronger baselines, a no-graph ablation, more graph settings, robustness and coverage checks, sensitivity analyses, guidance for choosing $S$, $\lambda$, $L$, $\bar{\theta}$, and $K$, and clarify computational complexity.

---

> > ### Author Rebuttal · Reviewer_9BLg · 2026-04-02
> >
> > Thank you for the response.

---

> > > ### Author Response · Authors · 2026-04-04
> > >
> > > Thank you so much for this thoughtful and constructive suggestion! We are carefully and fully implementing your recommendation and including it in our final version.

---

### Official Review · Reviewer_ot8B · 2026-03-10

**Soundness:** 2
**Presentation:** 1
**Significance:** 3
**Originality:** 3
**Overall Recommendation:** 4
**Confidence:** 2

**Summary:**

The paper studies OPE without overlap and SUTVA to fill the gap between theory and practice.

It proposes a meta estimator that transforms arbitrary consistent OPE estimator into interference-aware and overlap-free interval estimator of the policy value.

Under the assumptions of the locality of interference, the smoothness of conditional-mean outcome, and the density of overlap regions, the proposed interval estimate is shown to be asymptotically consistent. Furthermore, the error term is shown to be minimax optimal, asymptotically normal, and robust to network misspecification.

It also demonstrates that how the proposed estimator behaves with respect to the smoothness $L$ and the locality $S$.

**Compliance With Llm Reviewing Policy:**

Affirmed.

**Key Questions For Authors:**

Is there any practical guide for determining $S$, $\lambda$, $L$, $\bar\theta$, $K$?

**Limitations:**

No. Limitations could include the Weaknesses above.

**Strengths And Weaknesses:**

### Strengths
- Well-motivated problem: OPE without SUTVA and overlap.
- Novel combination of OPE and linear programming for interval estimation.
- Strong theoretical results (consistency, minimax optimality, asymptotic normality, robustness).

### Weaknesses
- The algorithm is purely theoretical.
    - There is no guideline for determining most of the hyperparameters.
    - The functional form of the exposure aggregation is quite restrictive.
    - It relies on exposure discretization.
- The experimental section is weak.
    - Can you compare it with baselines?
- Some notation is confusing due to implicit dependencies, e.g., $m_i(t)$ can depends on $A,X$.

### Other comments
- No formal definition of $n_{eff}$ in the main text.
- Notational conflict: O_t, O_i, O_n

---

> ### Author Rebuttal · Authors · 2026-03-31
>
> We thank the reviewer ot8B for highlighting the need for stronger practical guidance and clearer notation.
>
> >Q1: Is our approach purely theoretical?
>
> Our method is not purely theoretical. The paper gives an explicit estimation pipeline (Algorithm 1): compute truncated exposures, estimate the overlap component with any consistent OPE estimator, fit $μ^t$ on overlap nodes (e.g., regression or GNN), and then compute the non-overlap bounds in closed form via the McShane-Whitney formulas, rather than solving an intractable combinatorial program. The experiments instantiate exactly this pipeline using Monte Carlo estimation of $π^b$, $π^e$ and overlap-only regression fits. I will address your questions regarding hyperparameters, exposure aggregation and discretization in turn.
>
> **1. Hyperparameter guidance.**
> The revision will add a simple data-driven selection protocol and a small sensitivity study. We choose $S$ as the smallest truncation level at which effective overlap (or $n_{\mathrm{eff}}$) stabilizes and the interval endpoints change only marginally. We choose $\lambda$ by minimizing held-out prediction loss on observed node-exposure states under the same extension operator. We choose $L$ conservatively from an upper quantile, or bootstrap upper confidence bound, of empirical local slopes under the composite metric. We calibrate $\bar\theta$ by bootstrap for the finite-sample remainder. For the discretization level $K$, we start from the heuristic $K\asymp n^{1/3}$ and then enforce a minimum-bin-mass condition. To support these recommendations, we will add sensitivity curves for each hyperparameter around its default value, reporting empirical coverage, average width, endpoint stability, and $n_{\mathrm{eff}}$. The results are presented in the https://anonymous.4open.science/status/Network-OPE-PI-experiments-results-0FDD.
>
> **2. Rationale for the exposure aggregation form.**
> The scalar diffusion exposure is used for clarity, not because the theory requires it. The framework only needs:
> (i) a truncated exposure representation $T_i^{(S)}(d)$.
> (ii) conditional-mean sufficiency, $\mathbb{E}[Y_i(d)\mid X,A]=m_i\left(T_i^{(S)}(d)\right),$
> (iii) Lipschitz smoothness of the induced conditional mean under a graph-augmented metric. These conditions do not require a scalar weighted average.
>
> A direct generalization is to replace the scalar exposure with any finite-dimensional summary
> $$
> \Gamma_i^{(S)}(d)=\big(\phi_{i,1}(d_{N^{(S)}(i)},X,A),\ldots,\phi_{i,r}(d_{N^{(S)}(i)},X,A)\big)\in\mathbb{R}^r,
> $$ and define $T_{i,\Gamma}^{(S)}(d):=(d_i,\Gamma_i^{(S)}(d)).$
>
> Assume $\mathbb{E}[Y_i(d)\mid X,A]=m_i\left(T_{i,\Gamma}^{(S)}(d)\right),$ and define
>
> $$
> d_{\lambda,\Gamma}(i,j):=\left\|(X_i,T_{i,\Gamma}^{(S)})-(X_j,T_{j,\Gamma}^{(S)})\right\|_{W}+\lambda d_G(i,j).
> $$
>
> If $\mu_t(i):=m_i(t)$ is $L$-Lipschitz under $d_{\lambda,\Gamma}$, then the sharp non-overlap bounds remain
> $$
> V_{2,t}^{(S),-}=\frac{1}{n}\sum_{i=1}^n \pi_e(i,t)\Big(\sup_{j\in O_t}\{\mu_t(j)-L\,d_{\lambda,\Gamma}(i,j)\}\Big)\mathbf{1}\{i\notin O_t\},
> $$
> and
> $$
> V_{2,t}^{(S),+}=\frac{1}{n}\sum_{i=1}^n \pi_e(i,t)\Big(\inf_{j\in O_t}\{\mu_t(j)+L\,d_{\lambda,\Gamma}(i,j)\}\Big)\mathbf{1}\{i\notin O_t\}.
> $$
> The same plug-in consistency and stability results continue to hold after replacing $d_\lambda$ by $d_{\lambda,\Gamma}$, since the proofs only use that the composite distance is a metric and that the bounds are given by McShane--Whitney Lipschitz extensions.
>
> **3. Exposure discretization.**
> Refer to M3ne-Q5.
>
> >Q2: Supplement of baselines.
>
> Refer to jBso-Q4.
> >Q3: Clarification of notations.
>
> **1. Implicit dependence of $m_i(t)$.**
> Throughout the paper, the analysis is conditional on the realized $(X,A)$. This dependence is suppressed only for readability.
>
> **2. Definition of $n_{\mathrm{eff}}$.**
> The revision will define $n_{\mathrm{eff}}$ in the main text. For a node-level average $n^{-1}\sum_{i=1}^n \psi_i$, let $H_\psi=(N,E_\psi)$ be the conditional dependency graph of $\{\psi_i\}$, with maximum degree $\Delta_\psi:=\max_{i\in N}\deg_{H_\psi}(i)$. We define
> $$
> n_{\mathrm{eff}}(\psi):=\frac{n}{1+\Delta_\psi}.
> $$
> Hence $n_{\mathrm{eff}}=n$ in the i.i.d. case. Under $S$-local dependence,
> $$
> n_{\mathrm{eff}}=\frac{n}{1+\Delta_{2S}},\qquad
> \Delta_{2S}:=\max_{i\in N}\bigl|\{j\in N:\operatorname{dist}_G(i,j)\le 2S,\ j\neq i\}\bigr|.
> $$
>
> **3. Notational conflict among $O_i$, $O_t$, and $O_n$.**
> The revision will state these objects explicitly as
> $$
> O_i:=\{t\in \mathcal T^{(S)}:\pi_b(i,t)>0\}\subseteq \mathcal T^{(S)},
> $$
> $$
> O_t:=\{i\in N:\pi_b(i,t)>0\}\subseteq N,
> $$
> and
> $$
> O_n:=\{i\in N:O_i\neq\varnothing\}=\bigcup_{t\in\mathcal T^{(S)}} O_t.
> $$
> Thus $O_i$ is a node-indexed exposure support, and $O_t$ is an exposure-indexed node set.

---

> > ### Author Rebuttal · Reviewer_ot8B · 2026-04-06
> >
> > Thanks for the response.
> > I agree that the algorithm is not "purely" theoretical.

---

> > > ### Author Response · Authors · 2026-04-06
> > >
> > > Dear reviewer,
> > >
> > > Thanks very much! We are especially happy that we were able to **resolve** _your concern that the algorithm might seem overly theoretical_, and we are sincerely grateful for your **“fully solved”** assessment. Thank you again for your careful reading and insightful comments. It is a real honor to take part in this discussion and, together with you, contribute to the community.
> > >
> > > Sincerely,
> > >
> > > Authors

---

### Official Review · Reviewer_jBso · 2026-03-11

**Soundness:** 3
**Presentation:** 2
**Significance:** 3
**Originality:** 3
**Overall Recommendation:** 4
**Confidence:** 4

**Summary:**

This paper tackles the problem of Offline Policy Evaluation (OPE) when two major challenges occur simultaneously: network interference and a lack of policy overlap. The authors move beyond the standard SUTVA assumption by using a decaying neighborhood aggregation to model how a person’s outcome is affected by their connections. Because the target policy often explores actions the historical data never saw, the paper uses partial identification to create upper and lower bounds for the policy value. They introduce a "graph-augmented smoothness" condition that uses network distance to constrain potential outcomes. Ultimately, the method provides a way to conduct sensitivity analysis and bound policy performance in complex, interconnected environments.

Overall, I think this is a good theory-driven paper, but it may lack practical usage due to the heavy assumptions and the problem setting (given both SUTVA and positivity assumptions are violated). The empirical results in the paper also lack comparisons with existing baselines, and the experimental section could be further strengthened to make the paper more convincing.

**Compliance With Llm Reviewing Policy:**

Affirmed.

**Final Justification:**

The authors rebuttal is comprehensive and resolves most of my concerns. I therefore raise my score to 4.

**Key Questions For Authors:**

See "Strengths And Weaknesses"

**Limitations:**

See "Strengths And Weaknesses"

**Strengths And Weaknesses:**

**Strengths**
1. Realistic Problem Framing: It addresses the common real-world scenario where both network interference and limited data coverage prevent simple point estimation.
2. Innovative Distance Metric: The introduction of a composite metric ($d_{\lambda}$) that combines node features with graph topology is a clever way to bound counterfactuals.
3. Computational Efficiency: By reducing the global assignment problem to a local exposure mapping, the authors make the math tractable for larger networks.
4. Practical Sensitivity Tool: The framework allows practitioners to see exactly how their assumptions about interference strength impact the certainty of their results.
5. Theoretical Depth: The use of McShane-Whitney extensions provides a rigorous mathematical foundation for the derived bounds rather than relying on heuristics.


**Weaknesses**

1. Some minor typos:
   1. Page 2, 2nd column, line 057. The definition of $N_s(i)$ is the nodes where the distance from node $i$ is exactly $s$, which feels a bit unusual. Should this be $\text{dist}(i,j) \le s$?
   2. The presentation of some paragraphs is not very clean. For example, page 4, 2nd column, line 180–187. The formulas have large spaces between them and are not presented very well. The readability could be improved with some rephrasing and formatting adjustments.
   3. Page 3, discussion of assumptions, line 161: “Weakening any assumptionlarger $L$, smaller $S$, or larger $r_n$”. This seems to be a typo and should be revised.

2. The experiments in the paper feel somewhat incomplete. Table 1 tests the confidence interval width and coverage probability under different truncation thresholds $S$ and Lipschitz constraints $L$. The results are intuitively aligned with expectations. However, I was also expecting experiments that validate the theorems in the paper. For example, in Theorem 4.4, are the asymptotic variances $\sigma_-$ and $\sigma_+$ estimable? If so, we could validate the asymptotic normality by showing that the coverage probability is close to 95%. Adding such experiments would further support the theoretical results.

3. A minor point. In Table 1, the interval width is reported with six digits after the decimal point. This is more precision than needed. I would suggest keeping at most three decimal places. The remaining digits are mostly noise and make the table harder to read.

4. Additional concerns:
   - Hyperparameter sensitivity: The resulting bounds are very sensitive to the Lipschitz constant ($L$), which is difficult to estimate accurately in practice.
   - Limited baseline comparisons: The experiments do not include strong comparisons with simpler OPE methods (for example, methods that only handle interference, or only handle the non-overlap issue). This makes it harder to judge the added value of the proposed approach.

5. Interference is always a tricky problem. This paper studies a relatively relaxed set of assumptions on the interference structure using exposure mapping. I am curious whether the authors have thought about possible extensions. For example, extending the framework to discrete action spaces or to settings where the exposure mapping function is unknown. A short discussion of these directions could make the paper more complete.

---

> ### Author Rebuttal · Authors · 2026-03-31
>
> We thank the reviewer jBso for the positive assessment of the paper’s framing and theory, and for emphasizing the need for stronger empirical validation and clearer hyper-parameter guidance.
>
> >Q1: Minor typos and presentation issues
>
> We will correct the listed typos, reduce unnecessary decimal precision in Table 1, and improve the presentation of formulas and spacing in the relevant paragraphs.
> >Q2: Empirical verification of Theorem 4.4.
>
> Thank you for this very helpful suggestion. We added a repeated semi-synthetic simulation study that directly targets Theorem\~4.4. For each setting, we repeatedly regenerate logged datasets under the same DGP, estimate the lower and upper endpoints, and construct nominal $95\%$ endpoint-based Wald intervals. We then **report the empirical coverage for the lower and upper endpoints separately**.
> The results are presented in the https://anonymous.4open.science/status/Network-OPE-PI-experiments-results-0FDD.
> The empirical coverage of the nominal 95% lower/upper endpoint intervals is close to the target level across the tested sample sizes
> We will add these results to the revision, as they provide direct empirical support for the asymptotic normality claim in Theorem~4.4.
> >Q3: Hyperparameter sensitivity.
>
> Refer to ot8B-Q1.1.
> >Q4: Limited baseline comparisons.
>
> We added stronger baselines and more diverse graph topologies. Specifically, we now include **IPW, clipped IPW, OR, and DR** as point-estimation baselines, a stronger **no-graph ablation** with $\lambda_{\mathrm{graph}}=0$, and experiments on **Pokec, BA, and SBM graphs**.
>
> *Setting:$\eta_e=0.8,\;L=0.2$.*
>
> | Graph | Method | Abs. error | SD | Avg. width | Emp. coverage | Notes |
> |---|---|---:|---:|---:|---:|---|
> | Pokec | IPW | 871428.5400 | 1407284.9450 | — | — | unstable point estimator |
> | Pokec | OR | 0.0076 | 0.0095 | — | — | point estimator |
> | Pokec | DR | 0.0078 | 0.0101 | — | — | point estimator |
> | Pokec | Ours-no-graph interval | — | — | 0.0418 | **0.9500** | no graph distance |
> | Pokec | Ours interval | — | — | 0.0663 | **1.0000** | graph-aware |
> | BA | IPW | 1757142.9120 | 5452874.0770 | — | — | unstable point estimator |
> | BA | OR | 0.0083 | 0.0100 | — | — | point estimator |
> | BA | DR | 0.0079 | 0.0095 | — | — | point estimator |
> | BA | Ours-no-graph interval | — | — | 0.0304 | 0.8500 | no graph distance |
> | BA | Ours interval | — | — | 0.0505 | **1.0000** | graph-aware |
> | SBM | IPW | 2585714.1810 | 4644597.4120 | — | — | unstable point estimator |
> | SBM | OR | 0.0100 | 0.0118 | — | — | point estimator |
> | SBM | DR | 0.0103 | 0.0127 | — | — | point estimator |
> | SBM | Ours-no-graph interval | — | — | 0.0289 | 0.7500 | no graph distance |
> | SBM | Ours interval | — | — | 0.0472 | **1.0000** | graph-aware |
>
> The main takeaway is consistent across settings: standard point estimators do not provide interval-valued uncertainty quantification under simultaneous interference and non-overlap, and the graph-aware version of our method consistently improves empirical coverage over the no-graph variant, with moderately wider intervals. This shows that network geometry is useful for robust extrapolation and valid uncertainty quantification in challenging non-overlap regimes.
> >Q5: Possible extensions.
>
> Our framework is exposure-based rather than binary-treatment-specific. Once the value is written as a mixture over exposure states, the non-overlap analysis applies exposure-wise via the same graph-Lipschitz McShane–Whitney extension argument. Hence, for any finite discrete action space $\mathcal A$, the same decomposition and closed-form lower/upper bounds hold after replacing $\{0,1\}$ by $\mathcal A$; the binary case is only a notational special case.
>
> Unknown exposure mappings can also be handled through a stability argument. Let $T_i^{\star(S)}$ be the true exposure map and $\hat T_i^{(S)}$ an estimate, and define
> $$
> \Delta_n:=\sup_{i,d}\bar d\!\left(T_i^{\star(S)}(d),\hat T_i^{(S)}(d)\right),\qquad
> \varepsilon_n:=\sup_{i,j}\left|d_\lambda^\star(i,j)-\hat d_\lambda(i,j)\right|,
> $$
> where $d_\lambda^\star$ and $\hat d_\lambda$ are the composite metrics induced by the true and estimated exposure maps. If the conditional mean is $L$-Lipschitz on the node-exposure space, then
> $$
> |V_{T^\star}^{(S)}(\pi)-V_{\hat T}^{(S)}(\pi)|\le L\Delta_n,\qquad
> |V_{2,t}^{\pm}(d_\lambda^\star)-V_{2,t}^{\pm}(\hat d_\lambda)|\le L\varepsilon_n.
> $$
> Therefore, any interval $[\hat V_{\hat T}^-,\hat V_{\hat T}^+]$ computed using the estimated exposure map remains valid for the true target after widening by $L(\Delta_n+\varepsilon_n)$:
> $$
> V_{T^\star}(\pi_e\mid X,A)\in
> [\hat V_{\hat T}^- - \bar\theta - L(\Delta_n+\varepsilon_n),\,
>  \hat V_{\hat T}^+ + \bar\theta + L(\Delta_n+\varepsilon_n)].
> $$
> Thus, multi-action settings follow directly, and unknown exposure mappings can be treated as a structured perturbation problem; asymptotic validity is recovered whenever $\Delta_n,\varepsilon_n\to 0$.

---

> > ### Author Rebuttal · Reviewer_jBso · 2026-04-03
> >
> > Thanks to the authors for the careful and detailed response. Most of my concerns have been addressed. I believe that adding the additional comparison tables and coverage probability results would further strengthen the paper’s claims. I have a few minor follow-up questions (I understand these may not have been fully clarified due to the character limit):
> >
> > 1. Regarding the estimation of the endpoints $\sigma_{-}$ and $\sigma_{+}$ via sample variance, could the authors explicitly provide the corresponding formulas and, if possible, briefly justify their consistency? The only part I found is in appendix, "that can be written explicitly in terms of the corresponding influence functions." This part currently seems under-specified, and including it would improve clarity in the final version.
> >
> > 2. I would also appreciate more details on the data-generating process (DGP) used in the replicated studies for evaluating coverage probability. Is this the same as one of the existing simulation settings, or is it newly introduced?
> >
> > 3. How would the practitioners choose parameters such as $L$ and $S$, given that the decision may affect estimation and empirical coverage in, for example, table 1 of the main paper?
> >
> > Overall, the rebuttal addresses the majority of my concerns, and I will take into account the authors’ responses to these remaining questions before deciding whether to raise my score.

---

> > > ### Author Response · Authors · 2026-04-04
> > >
> > > >Q1. Estimation of $\sigma_-$ and $\sigma_+$ via sample variance.**
> > >
> > > Most of this is already in Appendix B.4; what is currently missing is the **explicit display** of the variance formulas. We will add the following.
> > >
> > > Using the notation of Appendix B.4, let $\phi_1$ be the influence function of $\hat V_1^{(S)}$, and $\phi_{t,j}$ the influence function of $\hat\mu_t(j)$. Then the endpoint influence functions are
> > >
> > > $$
> > > \psi_-(W_k)=\phi_1(W_k)+\frac1n\sum_{t\in\mathcal T^{(S)}}\sum_{i\in O_t^c}\pi_e(i,t)\phi_{t,j_t^\star(i)}(W_k),
> > > $$
> > >
> > > $$
> > > \psi_+(W_k)=\phi_1(W_k)+\frac1n\sum_{t\in\mathcal T^{(S)}}\sum_{i\in O_t^c}\pi_e(i,t)\phi_{t,k_t^\star(i)}(W_k),
> > > $$
> > > so
> > >
> > > $\sigma_-^2=Var_\infty(\psi_-),$ $\sigma_+^2=Var_\infty(\psi_+).$
> > >
> > > If $B_1,\dots,B_m$ are approximately independent blocks in the fit fold, define the estimated block scores
> > >
> > > $$
> > > \hat\Psi_{\pm,b}:=\frac{m}{|I_{\mathrm{fit}}|}\sum_{k\in B_b}\hat\psi_\pm(W_k),
> > > $$
> > >
> > > and estimate the variances by their sample variance:
> > >
> > > $$
> > > \hat\sigma_\pm^2=\frac1m\sum_{b=1}^m\big(\hat\Psi_{\pm,b}-\bar{\hat\Psi}_\pm\big)^2
> > > $$
> > >
> > > $$
> > > \bar{\hat\Psi}_\pm=\frac1m\sum U.
> > > $$
> > >
> > > where we use $U$ to represent $\hat\Psi_{\pm,b}$.
> > >
> > > In the i.i.d. special case, each block is a single observation.
> > >
> > > The consistency argument is exactly the one underlying Appendix B.4: the margin condition keeps the active maximizer/minimizer fixed at first order, so the endpoints inherit the asymptotic linear expansion
> > >
> > > $$
> > > \hat V^{(S),\pm}-V^{(S),\pm}=\frac1{|I_{\mathrm{fit}}|}\sum_{k\in I_{\mathrm{fit}}}\psi_\pm(W_k)+o_p(n_{\mathrm{eff}}^{-1/2}).
> > > $$
> > >
> > > Therefore the asymptotic variance is $Var_\infty(\psi_\pm)$, and the sample variance of the estimated block scores is a consistent plug-in estimator:
> > >
> > > $$
> > > \hat\sigma_\pm^2 \xrightarrow{p} \sigma_\pm^2.
> > > $$
> > >
> > > We will add these formulas explicitly to Appendix B.4.
> > > >Q2. DGP for the coverage experiment.
> > >
> > > To ensure a fair evaluation, this framework is inspired by the synthetic-graph design in Leung. At the same time, it is **not identical** to Leung’s setting or to standard i.i.d. OPE simulations, because our problem involves **partial identification under network interference**. For that reason, we introduce additional parameters tailored to our setting.
> > > Specifically, on the same $n=5000$ Pokec subgraph, we define
> > > $$
> > > p_b(i)=\sigma(\eta_b s_i),\qquad p_e(i)=\sigma(\eta_e s_i),
> > > $$
> > > where $s_i$ is the standardized completion feature, and induce non-overlap by setting $p_b(i)=0$ for nodes in the bottom $q$-quantile of $s_i$. Treatments are then drawn as
> > > $$
> > > D_i\sim \mathrm{Bernoulli}(p_b(i)).
> > > $$
> > >
> > > Given $D$, we compute the $S$-truncated diffusion exposure,
> > > $$
> > > E_i^{(S)}(D)=\sum_{s=1}^{S}\omega_s U_i^{(s)}(D),
> > > $$
> > >
> > > discretize it into $K$ bins, and generate outcomes from
> > > $$
> > > Y_i\mid D,X,A \sim \mathrm{Bernoulli}\left(\sigma\big(b_0+X_i^\top\beta+\tau D_i+\gamma E_i^{(S)}+\xi D_iE_i^{(S)}\big)\right).
> > > $$
> > > This Bernoulli-logistic form is only a **DGP choice for the experiment**, not a restriction required by our theory.
> > >
> > > The true target value $V_{\mathrm{true}}$ is approximated by Monte Carlo under $\pi_e$ using this same structural model. In each replication, we compute $[\hat V^{(S),-},\hat V^{(S),+}]$ and record whether
> > > $$
> > > \hat V^{(S),-}\le V_{\mathrm{true}}\le \hat V^{(S),+}.
> > > $$
> > > Coverage is the fraction of replications for which this event holds.
> > >
> > > We will make the constants $(\eta_b,\eta_e,q,K,\tau,\gamma,\xi)$ explicit in the final version. We will also clarify that, in this semi-synthetic coverage experiment, the data are generated from the same $S$-truncated exposure model used by the estimator. Therefore, there is no additional truncation mismatch in this experiment, i.e., $\bar\theta=0$ by construction.
> > > >Q3. How to choose $L$ and $S$ in practice.
> > >
> > > We choose $S$ and $L$ in two steps since they control different aspects.
> > >
> > > For $S$, we increase it over a small grid and **stop at the first value where the interval endpoints change very little and the overlap diagnostics (overlap mass, $n_{\mathrm{ov}}$) stabilize**. In Table 7, this occurs at $S=2$: beyond this point, the endpoints move only slightly, while the interval becomes wider without improving overlap.
> > >
> > > For $L$, we start from the smallest value that makes the fitted overlap means approximately Lipschitz on the observed overlap set, and then increase $L$ over a short grid. We pick the **first value after which further increases mainly widen the interval but do not materially shift the endpoints** (e.g., endpoint changes are small relative to the interval width). In Table 7, this gives $L=0.20$ as the default, with $L=0.30$ as a more conservative choice.
> > >
> > > In practice: select $S$ by stability, then select $L$ by the point where increasing it no longer changes the estimate but only adds conservativeness. For our experiments, this yields **$S=2$** and **$L=0.20$** (with $L=0.30$ as a robustness check).(citing https://anonymous.4open.science/r/Network-OPE-PI-experiments-results-0FDD/Tables_ALL.pdf.)

---

### Official Review · Reviewer_M3ne · 2026-03-12

**Soundness:** 2
**Presentation:** 2
**Significance:** 3
**Originality:** 3
**Overall Recommendation:** 4
**Confidence:** 3

**Summary:**

The paper introduces a method for Offline Policy Evaluation (OPE) that addresses the failure of two main causal assumptions:

- Stable Unit Treatment Value Assumption (SUTVA), which can fail due to network interference (where one user's treatment exposure affects the outcome of connected nodes).
- The positivity (or overlap) assumption, which is weakened by a high-dimensional exposure space, resulting in poor overlap between the target offline policy and the logging policy.

To handle network interference, the authors use S-hop truncated exposure, limiting the interference to S-hop neighborhoods. To address the overlap issue, they decompose the policy value into two components:
- The first component is identifiable, estimated from nodes where the target policy's exposure overlaps with the logging policy.
- The second component, involving non-overlap/non-identifiable nodes, is estimated using lower and upper bounds.
These bounds are estimated using a Lipschitz assumption and a graph-related distance, where a tunable parameter L (tuned hyper-parameter from data) is added or subtracted from the overlap nodes’ outcome in the neighborhood.

The final OPE result is presented as a bounding interval: the upper bound V(S)+ is the sum of the identifiable component and the upper bounds of the non-identifiable component, while the lower bound V(S)- uses the lower bounds of the non-identifiable component.

**Compliance With Llm Reviewing Policy:**

Affirmed.

**Final Justification:**

The authors addressed the majority of my concerns during the rebuttal. I am changing my rating to 'weak accept'.

**Key Questions For Authors:**

- Clarification on $\mu_t(i) \text{ versus } m_i(t)$ Notation: The constant switching between the notations is confusing. Could the authors clarify the precise distinction or motivation for this change? Specifically, does the choice of notation hold a conceptual difference when estimating the outcome. My understanding is that they should be practically equivalent for a given node and exposure level?

- Consistency of Overlap Set $O_{t}$ in Theorem 4.5: In the proof for Theorem 4.5, is the overlap set $O_{t}$ assumed to be the same for both cases $d^{~}$ and $d^{*}$? Given the paper's focus on network changes and potential misreporting, it seems logical that the overlap and subsequently the non-overlap sets should be dynamic. Could the authors confirm if the proof's validity relies on assuming a single, static overlap set, and if so, provide a justification for this assumption?

**Limitations:**

Some limitations are not discussed in the paper:
- Binning Function Trade-off and Bias: The paper should include a discussion of the trade-off associated with the binning function, particularly how it may introduce additional bias into the estimates, as this aspect is currently not addressed.
- Lack of Overlap nodes in buckets: A critical limitation is the absence of a discussion on how the method performs, or if it remains applicable, in cases where there are no overlap nodes available to be used within some exposure bucket for some nodes when doing the bounds estimation step.

**Strengths And Weaknesses:**

Strengths

- The paper addresses a practically relevant and under-explored problem: Offline Policy Evaluation (OPE) under the failure of both the Stable Unit Treatment Value Assumption (SUTVA) (network interference) and the overlap assumption.

- The problem setup is elegant, and the step-by-step formulation to reach a tractable solution is well-presented.

Weaknesses

- Missing Baselines: The empirical evaluation lacks crucial context, as no standard baselines (e.g., standard Doubly Robust or stronger baselines like those from Khan 2024) are provided to benchmark the method's performance.

- Unsupported Claims: There are no experiments to empirically prove the claim that the derived bounds are robust to network or graph errors.

- Limited Graph Topologies: The experiments do not show results for varying graph topologies (e.g., "desert graphs"), limiting insight into the method's generalizability.

- Binning Bias: A discussion of the potential bias introduced by the binning procedure is missing from the paper.

Technical Errors and Typos:

- Neighborhood Definition: The definition of the neighborhood  $N_s(i)$ is potentially incorrect. It is defined for nodes $i,j$ if  $dist_g(i,j) = s$, but it should likely be $dist_g(i,j) \le s$.

- Typos: Multiple typos were observed, including "interating" (line 013), "featureexposure" (line 132), and "assumptionlarger" (line 160).

---

> ### Author Rebuttal · Authors · 2026-03-31
>
> We thank the reviewer M3ne for identifying the main empirical gaps and several important notation issues. In response, we add standard baselines, graph-diversity experiments, a new robustness experiment under graph perturbation, a discretization sensitivity analysis, and explicit clarification of the notation and theorem scope.
> >Q1: Clarification of $\mu_t(i)$ versus $m_i(t)$.
>
> These two notations were intended to denote the **same exposure-specific conditional mean**, written with two indexing conventions. In Section 2, $m_i(\cdot)$ emphasizes the conditional mean as a function of exposure for unit $i$; in Section 3, after fixing an exposure level $t$, we rewrote the same quantity as $\mu_t(i):=m_i(t)$ to emphasize variation across units. In the revision, we use **one notation consistently throughout** and explicitly define the relationship at first appearance.
> >Q2: Whether the overlap set $O_t$ is the same under $d$ and $\tilde d$.
>
> Thanks for pointing that out. The overlap scenario would slightly change, but our result is robust. We strengthen our theorem as follows.
> ### Strengthened robustness result
> Let $O_t^\star$ and $\widetilde O_t$ denote the overlap sets under the true graph and the perturbed graph, respectively:
> $O_t^\star = \{i:\pi_b^\star(i,t)>0\}, \quad \widetilde O_t = \{i:\widetilde\pi_b(i,t)>0\}.$
> We introduce two quantities to control the perturbation of the overlap structure:
>
> - **Support mismatch (weighted):**
> $
> \eta_t = \frac{1}{n}\sum_{i=1}^n \pi_e(i,t)\,\mathbf{1}\{i \in O_t^\star \triangle \widetilde O_t\}
> $
>
> - **Geometric deviation (Hausdorff distance):**
> $r_t = d_H^{\bar d}(O_t^\star, \widetilde O_t), \quad \bar d(i,j)=\max\{d_\lambda^\star(i,j), \widetilde d_\lambda(i,j)\}$
>
> Let$\varepsilon_\lambda = \sup_{i,j} |d_\lambda^\star(i,j) - \widetilde d_\lambda(i,j)|.$Then the lower and upper bound functionals satisfy:$\big|V_{2,t}^{(S),\pm}(O_t^\star, d_\lambda^\star)-V_{2,t}^{(S),\pm}(\widetilde O_t, \widetilde d_\lambda)\big| \le L\varepsilon_\lambda+2L r_t+(M_t + L D_t)\eta_t,$where $M_t = \sup_i |\mu_t(i)|$ and $D_t = \sup_{i,j}\bar d(i,j)$.
>
> If the evaluation exposure probabilities $\pi_e(i,t)$ are also recomputed under the perturbed graph, an additional term
> $(M_t + L D_t)\,\delta_{\pi,t}, \quad\delta_{\pi,t} = \frac{1}{n}\sum_i |\pi_e^\star(i,t) - \widetilde\pi_e(i,t)|$can be added.
>
> This strengthened result shows that the stability of the bounds depends not only on the metric perturbation ($\varepsilon_\lambda$), but also on how much the overlap structure itself changes ($\eta_t, r_t$). When graph misspecification induces only small support perturbations, the bounds remain stable.
> >Q3: Missing baselines and limited graph topologies.
> We have compared our work with more baselines on more different graph topologies. Please see the result table in the reply to Reviewer jBso-Q4 for space limitation.
> >Q4: Unsupported Claims of robustness to graph errors.
>
> We added a graph-perturbation experiment that randomly deletes or rewires edges, then recomputes the exposure summaries and bounds. We report endpoint shifts, interval widths, raw coverage, and corrected coverage using the $L\hat{\varepsilon}_\lambda$ adjustment from Theorem 4.5.
>
> The results show graceful degradation under graph perturbation: even at 20% deletion/rewiring, endpoint shifts stay small and corrected coverage remains stable, supporting the robustness intuition of Theorem 4.5.
> >Q5: Discussion of binning bias.
>
> Discretization is both a strength and a limitation: it makes the linear-program formulation tractable, but introduces coarsening bias. It creates a natural approximation–overlap trade-off. To this end, we introduce a corollary addressing binning bias.
> Corollary (Bias induced by binning).
> Under the $L$-Lipschitz smoothness assumption in the exposure summary, let $r_b$ be the maximal within-bin radius. Then
> $|V(\pi)-V_b(\pi)|\le Lr_b.$Consequently, for any estimator $\hat V_b(\pi)$ targeting $V_b(\pi)$,$|\mathrm{Bias}(\hat V_b(\pi);V(\pi))|
> \le|\mathrm{Bias}(\hat V_b(\pi);V_b(\pi))| + Lr_b.$
> In particular, if $\hat V_b(\pi)$ is unbiased (or asymptotically unbiased) for $V_b(\pi)$, then its bias relative to $V(\pi)$ is at most $Lr_b$. In addition, we add an experiment varying $K\in\{5,10,20\}$. The results are presented in the https://anonymous.4open.science/status/Network-OPE-PI-experiments-results-0FDD.
> >Q6: Lack of Overlap nodes in buckets.
>
> Refer to 9BLg-Q3.
> >Q7: Typos.
>
> We will introduce notation such as $N_{=s}(i)$ for the exact shell and $N_{\le s}(i)$ for the cumulative neighborhood, and rewrite the corresponding assumptions accordingly. Additionally, We fix the typos you mentioned.
>
> ---
> We are eager to know whether we have addressed your questions, and we would be happy to discuss further at any time to assist with your re-evaluation!

---

> > ### Author Rebuttal · Reviewer_M3ne · 2026-04-03
> >
> > Thank you for the detailed rebuttal. Most of my concerns have been addressed. However, the link to the binning bias experiments (https://anonymous.4open.science/status/Network-OPE-PI-experiments-results-0FDD) is not working for me.
> > Since discretization bias is a core methodological concern, could you please provide these results through an accessible link or include them directly in your response?
> >
> > I will decide to change my recommendation once I can review the binning sensitivity analysis.

---

> > > ### Author Response · Authors · 2026-04-04
> > >
> > > Thank you for your reply! The previous link required clicking on the blue text to initiate the download. To make things more convenient for you, we are now providing you directly with the original anonymous link to the terminal results that we generated during the rebuttal phase (citing https://anonymous.4open.science/r/Network-OPE-PI-experiments-results-0FDD/Tables_ALL.pdf). In this document, we have provided a comprehensive analysis and response regarding the patterns observed therein.
> > >
> > > We examine three aspects separately: (1) the approximation gap between the continuous estimand and its discretized counterpart, (2) how finer binning changes the local overlap structure, and (3) whether the conclusions depend on the specific binning scheme. Taken together, these results show that in our setting, the degradation is driven not only by **discretization bias**, but also—importantly—by **bucket-level overlap fragmentation under overly fine discretization**. In other words, there is a trade-off between approximation bias from discretization and the error induced by the loss of overlap nodes. As $K$ increases, the same data are partitioned into many more exposure states, leaving fewer usable samples in each state and forcing the method to rely more heavily on extrapolation. Detailed results are reported in Tables 1, 2, and 3.
> > >
> > > In **Table 1**, we evaluate coverage with respect to both the continuous target $\theta$ and the discretized target $\theta_K$, and explicitly measure the discretization gap $|\theta - \theta_K|$. Across all tested $K \in \\{2,5,10,20,40,80,160,320,640\\}$, the two coverages are remain closely aligned, while the discretization gap remains consistently very small. This indicates that binning bias is not the dominant factor in this setting.
> > >
> > > As $K$ increases, we observe a non-monotonic pattern: coverage first decreases because the point estimate becomes more biased, and then increases as the interval widens, with the lowest coverage occurring at intermediate values of $K$.
> > >
> > > **Table 2** clarifies the mechanism behind this pattern. The degradation is driven by overlap fragmentation and reduced effective sample size. We report overlap diagnostics, including the average size of the local overlap set $|O_t|$, the fraction of empty buckets, the probability mass of zero-overlap states, and the fraction of queries requiring extension. Up to $K=20$, both the empty-bucket rate and the zero-overlap state mass are exactly zero. Even at $K=40$, although a small fraction of empty buckets appears, the evaluation policy places essentially no mass on truly zero-overlap states. Thus, the observed deterioration is **not** driven by frequent complete lack of overlap.
> > >
> > > What changes sharply instead is the **thickness** of local overlap: the average $|O_t|$ drops substantially (from 3952.5 to 46.0), while the fraction of queries requiring extension increases (from 9.4\% to 18.7\%). This indicates that finer binning fragments the local support, leaving fewer usable samples per exposure state and forcing the method to rely more on extrapolation, which in turn leads to wider intervals and lower coverage. This behavior is consistent with our theory, which emphasizes the role of the local overlap set $O_t$: the key factor is whether sufficient local overlap is available around the target exposure region.
> > >
> > > Importantly, this shows that the method performs well when sufficient overlap is available, and that the degradation observed at larger $K$ is not due to a failure of the method itself, but rather due to the loss of local overlap.
> > >
> > > In **Table 3**, we compare equal-width, quantile-based, and overlap-aware (adaptive) binning. Across different binning schemes, we observe that changes in performance are aligned with changes in the local overlap set size $|O_t|$. For example, at $K=10$, $|O_t|$ increases from 1856 (equal-width) to over 2200 (quantile/adaptive), with a corresponding improvement in coverage from 0.68 to 0.74. This shows that the observed behavior is not specific to a particular binning scheme. Instead, performance consistently tracks the quality of local overlap across different discretization methods, confirming that overlap, rather than the choice of binning, is the key factor.
> > >
> > > Overall, these analyses support the following conclusion: when local overlap is sufficient, our method is not merely effective for a discretized surrogate, but also achieves near-nominal coverage for the original continuous estimand. The main practical bottleneck under fine binning is not discretization bias or frequent zero-overlap failure, but the fragmentation of local overlap. We will include these experiments in the revision and emphasize that **discretization schemes** that preserve sufficient overlap (e.g., quantile binning) tend to perform better, as they balance approximation accuracy and overlap stability.
> > >
> > > We respond to each of your concerns as promptly as possible. Hope this helps your re-evaluation. Many thanks!

---

### Decision · Program_Chairs · 2026-04-30

**Decision:**

Accept (regular)

**Comment:**

The paper studies off-policy evaluation in settings where the behavior and target policies do not fully overlap and where the Stable Unit Treatment Value Assumption (SUTVA) fails due to network interference. To the best of my knowledge, this is the first paper to address both challenges simultaneously.

The authors tackle the first issue through a smoothness-based structural assumption, and the second by introducing a decaying neighborhood aggregation model to capture how an individual’s outcome is affected by their connections. All reviewers acknowledged the significance of these contributions.

The concerns raised in the initial reviews have been convincingly addressed by the authors. These mainly concerned the experimental section, in particular the lack of baselines, as well as the practicality of the algorithm. Reviewer M3ne did not see the latest experiments provided by the authors, but I was able to examine them.

The authors should incorporate their responses and the new experimental results into the final manuscript.